# CausalPFN: Amortized Causal Effect Estimation via In-Context Learning

**Vahid Balazadeh**[*12]   **Hamidreza Kamkari**[*3]   **Valentin Thomas**[3]   **Benson Li**[12]   **Junwei Ma**[3]
**Jesse C. Cresswell**[3]   **Rahul G. Krishnan**[12]
vahid@cs.toronto.edu, hamid@layer6.ai
[1] University of Toronto   [2] Vector Institute   [3] Layer 6 AI

## Abstract

Causal effect estimation from observational data is fundamental across various applications. However, selecting an appropriate estimator from dozens of specialized methods demands substantial manual effort and domain expertise. We present CausalPFN, a single transformer that *amortizes* this workflow: trained once on a large library of simulated data-generating processes that satisfy ignorability, it infers causal effects for new observational datasets out of the box. CausalPFN combines ideas from Bayesian causal inference with the large-scale training protocol of prior-fitted networks (PFNs), learning to map raw observations directly to causal effects without any task-specific adjustment. Our approach achieves superior average performance on heterogeneous and average treatment effect estimation benchmarks (IHDP, Lalonde, ACIC). Moreover, it shows competitive performance for real-world policy making on uplift modeling tasks. CausalPFN provides calibrated uncertainty estimates to support reliable decision-making based on Bayesian principles. This ready-to-use model requires no further training or tuning and takes a step toward automated causal inference (https://github.com/vdblm/CausalPFN).

## 1 Introduction

Causal inference—estimating the effects of interventions from data—is fundamental across numerous domains, including public policy, economics, and healthcare [75, 5, 47]. The central challenge lies in estimating causal quantities from observational data: records collected without explicit interventions, where confounding factors can obscure true causal effects. Various causal identification settings have emerged to address this challenge [45, 6, 10, 71]. Perhaps the most common one is to assume no unobserved confounding (ignorability or backdoor) [98, 87].

Even within the conceptually straightforward ignorability framework, researchers have developed dozens of specialized causal estimators over the past four decades. Prominent examples include Meta-Learners [57], doubly robust methods [30, 52], double machine learning

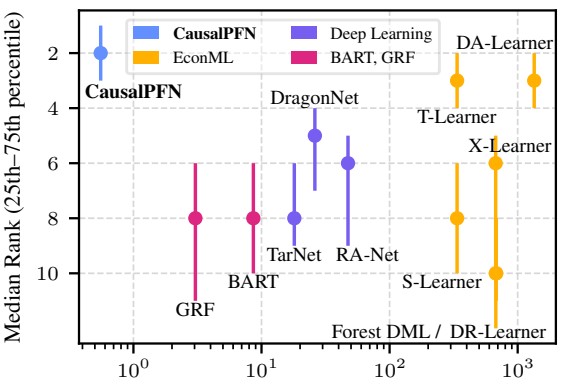

Figure 1: **Time vs. Performance.** Comparison across 310 causal inference tasks from IHDP, ACIC, and Lalonde. CausalPFN achieves the best average rank (by precision in estimation of heterogeneous effect) while being much faster than other baselines.

---

*Equal Contribution

39th Conference on Neural Information Processing Systems (NeurIPS 2025).

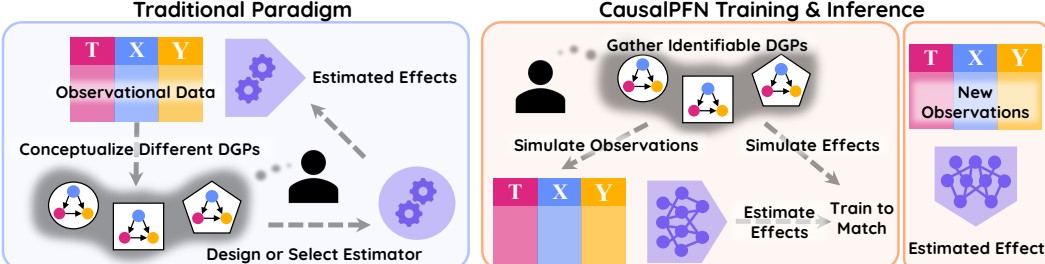

Figure 2: **Traditional Causal Inference vs. CausalPFN**. *(Left)*: A domain expert manually builds or selects an estimator for a DGP that they deem appropriate for the given data. *(Right)*: The domain expert simulates diverse DGPs for pre-training, and a transformer learns to amortize causal inference automatically.

(DML) [16, 29], and neural network approaches [104, 106, 19–22], among others [97, 56, 70, 65]. This large number of estimators creates practical challenges as domain expertise is required to select, tune, or design the most appropriate estimator for each application [107, 103, 27, 2, 81, 73].

The Bayesian paradigm offers an elegant framework to address these challenges [99, 46, 47, 40, 9]; rather than manually designing or selecting the best estimator, one can: (1) parameterize an appropriate prior distribution over plausible underlying causal mechanisms, i.e., the data-generating processes (DGPs), (2) define the causal estimand as a functional of the DGP parameters, (3) compute a posterior distribution over DGPs conditioned on observed data, and (4) derive the posterior predictive distribution (PPD) of the causal estimand. However, the practical adoption of Bayesian methods remains limited. Computing posterior distributions typically requires expensive sampling methods [84, 40], which often leads researchers to make specific assumptions about the DGPs or priors that are not necessarily reflective of the complexity of the downstream tasks [36, 62].

Meanwhile, an emerging area in deep in-context learning suggests using large models that can approximate PPDs by taking the entire list of observations as context and amortize the expensive process of posterior inference [32, 31, 54]. A successful example is the prior-fitted network (PFN) [80] that achieved remarkable performance in tabular prediction tasks [42, 69, 37, 43, 118, 76, 66]. PFNs employ transformer architectures trained on large-scale simulated DGPs, representing a rich prior, to perform posterior predictive inference via in-context learning; given a dataset of input-output examples as context, they can predict outputs for new inputs. PFNs shift the computational burden from inference time to (pre-)training, producing a single set of model parameters that can make fast and accurate predictions on unseen datasets. However, they are only designed for regression and classification, not for causal inference.

We propose to bridge the large-scale training of amortized models with Bayesian causal inference and introduce CausalPFN, a transformer model for causal effect estimation via in-context learning. Our framework leverages a general-purpose prior, based on the *ignorability* assumption, to generate a vast collection of simulated DGPs. By training on these diverse DGPs, our method learns to infer the causal estimands directly from observational data. While our approach requires an expensive pre-training phase, once complete it is ready for inference on new datasets with no further training, fine-tuning, or hyperparameter optimization. Hence, CausalPFN is easy-to-use, efficient for inference, and shows remarkably strong performance as an estimator. Figure 1 illustrates the relative performance and efficiency of our method compared to standard baselines. For inference on an unseen dataset, CausalPFN requires only forward passes, whereas baseline methods have additional costs including hyperparameter tuning or cross-validation. We therefore report the computational time for all of these stages for the baselines to reflect the total costs of predicting on a new dataset.

We show CausalPFN's workflow compared to traditional causal inference in Figure 2. Our key contributions are: *(i)* To our knowledge, for the first time, we demonstrate that a single transformer-based model trained on a diverse library of simulated DGPs can match or surpass specialized estimators across multiple datasets without task-specific tuning. Specifically, CausalPFN achieves the best average rank on CATE across IHDP, ACIC, and Lalonde, and competitive ATE performance, without task-specific tuning. *(ii)* We highlight CausalPFN's competitive out-of-the-box performance for real-world policy making on various uplift modeling tasks. *(iii)* We theoretically characterize the assumptions under which CausalPFN's estimates are asymptotically consistent. *(iv)* We develop a principled uncertainty quantification framework for CausalPFN to produce finite-sample calibrated credible intervals for the estimates. *(v)* Finally, we release our model's weights with a user-friendly API, streamlining the adoption of CausalPFN as a capable estimator. CausalPFN is fast, ready-to-use, and does not require any further training or hyperparameter tuning.

## 2   Background

**Causal Effect Estimation.** We adopt the potential–outcomes framework for causal inference [100]. Let $T \in \mathcal{T}$ denote the treatment from a finite treatment set $\mathcal{T}$, and $\mathbf{X} \in \mathcal{X}$ the observed covariates. For every $t \in \mathcal{T}$, $Y_t \in \mathbb{R}$ is the potential outcome under treatment $t$, while the observed (factual) outcome is $Y := Y_T$. We call the joint distribution $P(\mathbf{X}, T, \{Y_t\}_{t \in \mathcal{T}}, Y)$ the *data-generating process* (DGP), and denote by $P_{\text{obs}}$ the marginal distribution over observed triples $(\mathbf{X}, T, Y)$. Given samples from $P_{\text{obs}}$, a central goal is to recover the *conditional expected potential outcomes* (CEPOs):

$$\mu_t(\mathbf{x}) := \mathbb{E}[Y_t \mid \mathbf{X} = \mathbf{x}], \qquad \forall t \in \mathcal{T}, \ \mathbf{x} \in \mathcal{X}. \tag{1}$$

For binary treatments, two common estimands, average treatment effect (ATE), and conditional average treatment effect (CATE) follow directly from the CEPOs. We refer to CEPOs, CATE, and ATE collectively as *causal effects*.

$$\text{ATE}: \quad \lambda := \mathbb{E}[Y_1 - Y_0] = \mathbb{E}[\mu_1(\mathbf{X}) - \mu_0(\mathbf{X})], \tag{2}$$
$$\text{CATE}: \quad \tau(\mathbf{x}) := \mathbb{E}[Y_1 - Y_0 \mid \mathbf{X} = \mathbf{x}] = \mu_1(\mathbf{x}) - \mu_0(\mathbf{x}). \tag{3}$$

Estimating causal effects from observational data is impossible without further assumptions: different DGPs can induce the same $P_{\text{obs}}$ but have different causal effects [87, 39, 47]. We thus define:

**Definition 1** (CEPO-Identifiability). For each $t \in \mathcal{T}$, CEPO-identifiability holds when $\mu_t$ can be written as a functional of the observational distribution $P_{\text{obs}}$.

Throughout, we assume *strong ignorability*, a standard *sufficient* assumption that makes CEPOs identifiable. Strong ignorability posits that, conditional on observed covariates, treatment assignment has positive probability for all $t \in \mathcal{T}$ and is independent of all potential outcomes [98, 97, 89]:

**Assumption 1** (Strong Ignorability). *(i)* $Y_t \perp\!\!\!\perp T \mid \mathbf{X}$ for all $t \in \mathcal{T}$ (Unconfoundedness), and *(ii)* $P(T = t \mid \mathbf{X}) > 0$ a.e. for all $t \in \mathcal{T}$ (Positivity).

**Bayesian Causal Inference.** A Bayesian formulation of causal inference considers an explicit likelihood model for the DGP [99, 84, 62]. Let $\psi$ be the parameter that indexes the DGPs $P^\psi(\mathbf{X}, T, \{Y_t\}_{t \in \mathcal{T}}, Y)$. A prior $\pi(\psi)$ encodes domain knowledge on parameters $\psi$. Given i.i.d. observations $\mathcal{D}_{\text{obs}} = \left\{ (\mathbf{x}^{(n)}, t^{(n)}, y^{(n)}) \right\}_{n=1}^N$ coming from the observational distribution $P_{\text{obs}}^\psi$, Bayes' rule yields the posterior $\pi(\psi \mid \mathcal{D}_{\text{obs}})$. For any functional $g(\psi)$—for example $g(\psi) = \mathbb{E}^\psi[Y_1 - Y_0]$ for ATE—the posterior predictive distribution (PPD)

$$\pi^g(\cdot \mid \mathcal{D}_{\text{obs}}) := \left[ B \mapsto \int \mathbb{I}(g(\psi) \in B) \, \pi(\psi \mid \mathcal{D}_{\text{obs}}) \, \mathrm{d}\psi \right], \qquad B \in \mathcal{B}, \tag{4}$$

is induced by the posterior distribution $\pi(\psi \mid \mathcal{D}_{\text{obs}})$ ($\mathcal{B}$ denotes the Borel $\sigma$-algebra over $\mathbb{R}$). Point estimates (posterior means) and credible intervals therefore arise automatically from these induced posteriors. Because the posterior is rarely available in closed form, one resorts to approximate inference such as Markov-chain Monte-Carlo (MCMC) [40] or variational inference [68, 48]. Such techniques have been applied with flexible priors including nonparametric BART models [40, 36], Dirichlet processes [64] and Gaussian processes [3]. In summary, the Bayesian paradigm offers a unified framework for inference on causal estimands and provides automatic uncertainty quantification.

**Amortizing Posterior Predictive Inference with Prior-Fitted Networks.** Running a new posterior inference for every dataset is computationally demanding, especially with high-dimensional covariates [36, 62]. Recent work shows that in-context transformers can *amortize* Bayesian prediction: instead of sampling from the posterior at test time, a single network is trained to map a context set directly to the PPD [31, 32, 54, 80, 37]. PFNs instantiate this idea for supervised learning [42].

Consider a supervised dataset $\mathcal{D}^{\text{SL}} = \{(\mathbf{x}^{(n)}, y^{(n)})\}_{n=1}^N$ and a prior $\pi^{\text{SL}}$ on parameters $\phi$ indexing $P^\phi(\mathbf{X}, Y)$. The Bayesian approach to predict the output for a new input $\mathbf{x}$ is to use the PPD

$$\text{PPD}(Y \mid \mathbf{X} = \mathbf{x}, \mathcal{D}^{\text{SL}}) := \int P^\phi(Y \mid \mathbf{X} = \mathbf{x}) \pi^{\text{SL}}(\phi \mid \mathcal{D}^{\text{SL}}) \, \mathrm{d}\phi. \tag{5}$$

Rather than approximating the posterior distribution $\pi^{\text{SL}}(\phi \mid \mathcal{D}^{\text{SL}})$ with MCMC or variational inference [49, 4, 82], PFNs directly parameterize the PPD using a single transformer model $q_\theta(Y \mid \mathbf{X}, \mathcal{D}^{\text{SL}})$ by minimizing the *data-prior loss*

$$\ell_\theta := \mathbb{E}_{\phi \sim \pi^{\text{SL}}, \, \mathcal{D}^{\text{SL}} \cup \{\mathbf{X}, Y\} \sim P^\phi} \left[ -\log q_\theta(Y \mid \mathbf{X}, \mathcal{D}^{\text{SL}}) \right]. \tag{6}$$

Crucially, training requires only *prior* samples $(\phi, \mathcal{D}^{\mathrm{SL}})$; no posterior sampling is needed. With a suitably rich prior, a single PFN can be applied *off-the-shelf* to diverse predictive problems [69, 43].

# 3    The Mathematical Framework of CausalPFN

Our primary estimands of interest are the CEPOs from (1). As shown in (2) and (3), CEPOs directly enable estimation of both ATE and CATEs. Therefore, we focus on developing an estimator that can accurately infer these quantities from the observational data. Specifically, we follow the Bayesian paradigm for causal inference, as introduced in Section 2, and parameterize CEPOs as $\mu_t(\mathbf{x} \; ; \; \psi)$. Given a suitably rich prior distribution $\pi$ over the DGPs, which we will explicitly design in Section 4, we define our target as the posterior predictive distribution of CEPOs:

**Definition 2** (CEPO-PPD)**.**  For each $t \in \mathcal{T}$ and covariate vector $\mathbf{x}$, the *CEPO-PPD* is

$$\pi^{\mu_t}(\cdot \mid \mathbf{x}, \mathcal{D}_{\mathrm{obs}}) \; := \; \left[ B \mapsto \int \mathbb{I}(\mu_t(\mathbf{x} \; ; \; \psi) \in B)\, \pi(\psi \mid \mathcal{D}_{\mathrm{obs}})\, \mathrm{d}\psi \right], \qquad B \in \mathcal{B}. \tag{7}$$

**Consistent Estimation of CEPOs.** The CEPO-PPD captures the epistemic uncertainty about the CEPO encoded in the posterior. A concentrated distribution $\pi^{\mu_t}$ indicates that the observations $\mathcal{D}_{\mathrm{obs}}$ are informative and enough samples are available to accurately pin down the true CEPO, whereas a high-variance distribution implies that the data is insufficient for estimation. With that in mind, we now study under which conditions increasing the size of the observations $\mathcal{D}_{\mathrm{obs}}$ allows us to accurately recover the true CEPO from the CEPO-PPD. This is given through the following informal result (re-stated and proven formally in Appendix B) which provides necessary and sufficient conditions on the prior $\pi$ under which the CEPO-PPDs enable consistent estimation of the CEPOs:

**Proposition 1** (Informal)**.**  *Under mild regularity assumptions, for almost all $\psi^{\star} \sim \pi$ and any set of i.i.d. samples $\mathcal{D}_{\mathrm{obs}} \sim P_{\mathrm{obs}}^{\psi^{\star}}$, we have that as $|\mathcal{D}_{\mathrm{obs}}| \to \infty$,*

$$\mathbb{E}_{\mu \sim \pi^{\mu_t}(\cdot \mid \mathbf{x}, \mathcal{D}_{\mathrm{obs}})}[\mu] \xrightarrow{a.s.} \mu_t(\mathbf{x} \; ; \; \psi^{\star}), \quad \forall t \in \mathcal{T}, \text{ and almost all } \mathbf{x} \in \mathcal{X}, \tag{8}$$

*if and only if the prior $\pi$ is CEPO-identifiable, that is for almost all $\psi \sim \pi$, the CEPOs $\mu_t(\cdot \; ; \; \psi)$ only depend on the observational distribution $P_{\mathrm{obs}}^{\psi}$ (Definition 1).*

*(Proof sketch)* We group all DGPs $\psi$ that share the same observational distribution $P_{\mathrm{obs}}^{\psi}$ into an equivalence class and induce a prior obtained from $\pi$ on the resulting quotient space. By Doob's theorem [26]–a classical result from Bayesian consistency theory–the posterior on this new prior almost surely concentrates on the true equivalence class once asymptotically many observations are given. Consequently, for any functional of the observations that is constant within each equivalence class, its posterior predictive converges almost surely to its true value. Importantly, the causal functional of interest, $\mu_t$, can be written as a functional of the observations if and only if the corresponding DGP has identifiable CEPOs. Thus, identifiability is both necessary and sufficient to ensure that $\mu_t$ is constant throughout the equivalence class, and for the consistency result to hold.

*(Remark 1)* While the algorithms in our paper use *strong ignorability*, Proposition 1 itself is an entirely general result and can be extended to DGPs that are not necessarily ignorable, but whose CEPOs satisfy identifiability in Definition 1. Importantly for our practical setting, when the prior $\pi$ enforces strong ignorability, Proposition 1 suggests that the CEPO-PPDs consistently recover the true CEPO.

*(Remark 2)* Proposition 1 highlights two key design principles for the prior $\pi$: $(i)$ $\pi$ must rule out non-identifiable cases, and, once identifiability is secured, $(ii)$ broadening $\pi$ increases the chance that a particular $\psi^{\star}$ lies within its support, thus enabling consistent recovery of the true CEPO for that $\psi^{\star}$.

**Learning the CEPO-PPD.** Having shown that CEPO-PPDs are useful for estimating the true CEPOs, we now describe how to learn them. Inspired by PFNs, we train a single transformer $q_{\theta}$ to approximate the full predictive distribution $\pi^{\mu_t}$. To fit this model, we introduce the following loss:

**Definition 3** (Causal Data-Prior Loss)**.**  For any $t \in \mathcal{T}$, we define the causal data-prior loss as

$$\mathcal{L}_t(\theta) \; := \; \mathbb{E}_{\psi \sim \pi, \, \mathcal{D}_{\mathrm{obs}} \cup \{\mathbf{x}\} \sim P_{\mathrm{obs}}^{\psi}} [-\log q_{\theta}(\mu_t(\mathbf{x} \; ; \; \psi) \mid \mathbf{x}, t, \mathcal{D}_{\mathrm{obs}})]. \tag{9}$$

In Appendix C, we show that minimizing $\mathcal{L}_t(\theta)$ also minimizes the KL-divergence between the true CEPO-PPD and $q_{\theta}$, leading to $q_{\theta}(\cdot \mid \mathbf{x}, t, \mathcal{D}_{\mathrm{obs}}) \approx \pi^{\mu_t}(\cdot \mid \mathbf{x}, \mathcal{D}_{\mathrm{obs}})$ for all $t \in \mathcal{T}$. This entire training

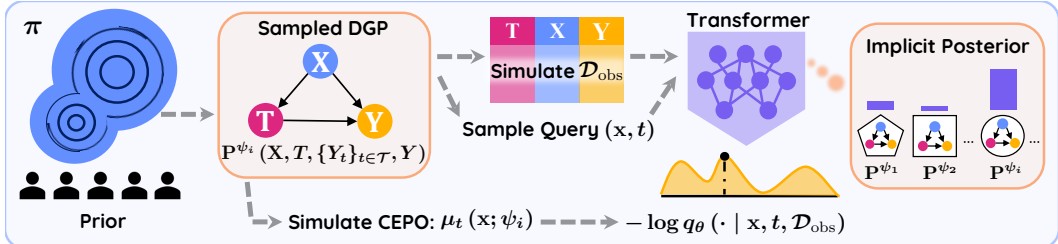

Figure 3: **Causal Data-Prior Training**. At each iteration an index $\psi_i \sim \pi$ is sampled *(left)*, yielding the DGP $P^{\psi_i}(\mathbf{X}, T, \{Y_t\}_{t \in \mathcal{T}}, Y)$. From this DGP we simulate an observational context $\mathcal{D}_{\text{obs}}$ and a query $(\mathbf{x}, t)$ with its true $\mu_t(\mathbf{x} ; \psi_i)$ *(center)*. Passing $(\mathbf{x}, t, \mathcal{D}_{\text{obs}})$ through the transformer predicts the CEPO-PPD $q_\theta(\cdot \mid \mathbf{x}, t, \mathcal{D}_{\text{obs}})$ *(in yellow)*, which is derived from an implicit posterior $\pi(\cdot \mid \mathcal{D}_{\text{obs}})$ that is *never* explicitly computed *(right)*. We train $\theta$ to minimize the causal data-prior loss *(bottom)*.

process shifts the computational burden from inference to pre-training: rather than evaluating the posterior $\pi(\psi \mid \mathcal{D}_{\text{obs}})$ at test time, the model learns to map observational data directly to the corresponding predictive distribution. When the model is well-fitted, the prior satisfies the assumptions of Proposition 1, and $\mathcal{D}_{\text{obs}}$ is sufficiently large, the predicted $q_\theta$ accurately pins down the true CEPO.

Figure 3 visually illustrates optimizing the causal data-prior loss using stochastic gradient descent: at each iteration, we sample a DGP $\psi_i \sim \pi$, generate an observational dataset $\mathcal{D}_{\text{obs}}$ from this DGP, and select a query point $(\mathbf{x}, t)$. We compute (simulate) the ground-truth CEPO $\mu_t(\mathbf{x} ; \psi_i)$ and feed both the observational data and query to the model. The model outputs a CEPO-PPD, and we update $\theta$ using gradient descent to increase the probability assigned to the true CEPO value. Through training, $\theta$ minimizes the data-prior loss and implicitly learns to perform posterior predictive inference, and estimate the predictive distribution $\pi^{\mu_t}$, without ever explicitly computing the posterior.

**Point & Distributional Estimation of Causal Effects.** Given observational data $\mathcal{D}_{\text{obs}}$ from an underlying $\psi^\star$, a natural point estimate for CEPOs is the mean of the predicted CEPO-PPD, $\mathbb{E}_{\mu \sim q_\theta(\cdot \mid \mathbf{x}, t, \mathcal{D}_{\text{obs}})}[\mu] \approx \mu_t(\mathbf{x} ; \psi^\star)$. These CEPO estimates can also form point estimates for CATEs using (3), and for ATEs using (2) by empirical averaging across units in $\mathcal{D}_{\text{obs}}$.

Beyond point estimation, the estimated CEPO-PPDs can also capture the epistemic uncertainty about the causal effects. We can use $q_\theta$ to construct credible intervals around CEPOs, CATEs, and ATEs via sampling from $q_\theta(\cdot \mid \mathbf{x}, t = 1, \mathcal{D}_{\text{obs}})$ and $q_\theta(\cdot \mid \mathbf{x}, t = 0, \mathcal{D}_{\text{obs}})$. We can then use these intervals to quantify the uncertainty of our estimated causal effects.

# 4 Implementing CausalPFN

While Section 3 presents the framework in general form (arbitrary finite $\mathcal{T}$ and identifiability), for implementation we focus on binary treatments $\mathcal{T} = \{0, 1\}$ under strong ignorability. These assumptions reflect the most common settings encountered by practitioners and serve as a natural starting point. Extending the implementation and algorithms to more general settings is left for future work.

**A Scalable Prior.** Here, we focus on designing an appropriate prior $\pi$ over DGPs that satisfies the theoretical requirements established in Proposition 1. This prior must balance two factors: First, it should contain a rich set of DGPs with sufficient coverage to approximate real-world scenarios—similar to the priors used in successful tabular predictive models like TabPFN [42, 43], TabDPT [69], and TabICL [92]. Second, and uniquely for causal inference, all DGPs in our prior must satisfy strong ignorability which

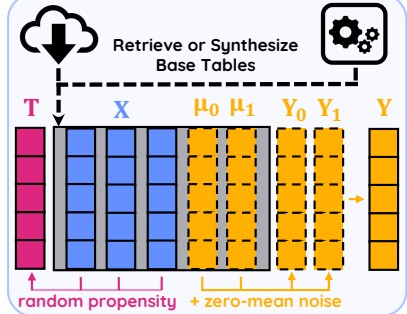

Figure 4: **Prior construction**. Sample diverse base tables (OpenML or synthetic TabPFN), select covariates $X$, draw treatment $T$ with a random propensity model, select columns $\mu_0, \mu_1$ and add zero-mean noise to form $Y_0, Y_1$, and $Y$.

directly implies identifiability of the prior. Moreover, the generated DGPs must allow us to access the ground-truth CEPOs, as required by the causal data-prior loss in Definition 3 for training.

To address these requirements, we develop a procedure that can transform *any* base table from standard tabular priors into a valid causal dataset, illustrated by Figure 4: *(i)* retrieve a base table

with $N$ rows from either a large library of tabular data[2] or synthesize it (details in Appendix D.1); *(ii)* randomly select columns with a varying number of covariates as $\mathbf{X}$; *(iii)* pick two other columns, relabel them as $\mu_0(\mathbf{X}), \mu_1(\mathbf{X})$; *(iv)* optionally add zero-mean noise to $\mu_0(\mathbf{X})$ and $\mu_1(\mathbf{X})$ to obtain $Y_0$ and $Y_1$, or simply set $Y_0 = \mu_0(\mathbf{X})$ and $Y_1 = \mu_1(\mathbf{X})$; these four steps simulate samples from the joint distribution $(\mathbf{X}, Y_0, Y_1)$; *(v)* generate a random function $f$, leveraging similar synthetic functions as in Hollmann et al. [42] to map covariates to their treatment logits; *(vi)* sample binary treatments $T \sim \text{Bernoulli}(\text{Sigmoid}(f(\mathbf{X})))$; *(vii)* finally, form the observed outcomes $Y := Y_T$.

The procedure above "simulates" a collection $\{t^{(n)}, \mathbf{x}^{(n)}, \mu_0^{(n)}, \mu_1^{(n)}, y^{(n)}\}_{n=1}^N$ from an underlying DGP that can be used to sample the observational data and obtain CEPOs necessary for training (recall Figure 3). This approach guarantees strong ignorability *by design*: since treatment $T$ is determined solely from $\mathbf{X}$, it is conditionally independent from the potential outcomes $Y_0, Y_1$. Additionally, by applying the sigmoid function, we ensure $0 < P(T = 1 \mid \mathbf{X}) < 1$, satisfying positivity. While this procedure primarily targets binary treatments, it can naturally extend to finite discrete treatments.

For the diversity aspect of $\pi$, we rely on the empirical success of existing tabular foundation models and the deliberate design in our generation process. Sampling covariates directly from a mix of real and synthetic tables yields data that is more likely to reflect the scenarios the model will face at inference. We assume no distributional assumptions on covariates and potential outcomes. Appendix D.1 details additional mechanisms for controlling treatment effect heterogeneity and positivity in our synthetic DGPs, as well as the detailed configurations of the prior-generation process.

**Model Architecture & Parallel Training.** We model $q_\theta$ using a PFN-style transformer encoder that receives a sequence of row tokens as *context* (i.e., $\mathcal{D}_{\text{obs}}$), where each token embeds a triplet $(t^{(n)}, \mathbf{x}^{(n)}, y^{(n)})$. At every iteration, we embed $B_Q$ batched *query* tokens $(t, \mathbf{x})$. We then apply 20 layers of self-attention and MLP layers, followed by a final projection layer to get $q_\theta(\cdot \mid \mathbf{x}, t, \mathcal{D}_{\text{obs}})$ for all the $(t, \mathbf{x})$ pairs in the batched query. The transformer uses the asymmetric masking used in PFNs: both context and query tokens attend only to the context tokens, ensuring that the predicted CEPO-PPDs are mutually independent.

To model each CEPO-PPD, we approximate it with a quantized histogram. We discretize the outcome axis into $L = 1024$ bins and let the network project the query tokens into $L$ logits. We then apply SoftMax to turn the logits into a quantized distribution $q_\theta(\cdot \mid \mathbf{x}, t, \mathcal{D}_{\text{obs}})[\ell], \forall \ell \in [L]$. At each round of gradient update, we place a Gaussian with a small $\sigma$ at the true CEPO $\mu_t(\mathbf{x})$ and integrate it over bins to obtain Gaussian quantized probabilities $\mathcal{N}(\mu_t(\mathbf{x}), \sigma^2)[\ell]$ and minimize the *histogram loss*:

$$\text{HL}[\mu_t(\mathbf{x}) \parallel q_\theta] = -\sum_{\ell=1}^L \mathcal{N}(\mu_t(\mathbf{x}), \sigma^2)[\ell] \cdot \log q_\theta[\ell]. \tag{10}$$

This loss is an approximation to the causal data-prior loss in (9); it coincides in the limit $\sigma \to 0$ and $L \to \infty$. The histogram loss formulation affords a tractable proxy for the continuous CEPO-PPD.

A more detailed overview of the architecture and procedures for point and interval estimates is illustrated in Figure 5; further details (e.g., parameter counts, compute, inference-time techniques, number of prior datasets, scalability, and speed) are available in Appendices D.2, D.3, and D.4.

## 5 Experiments

**Baseline Causal Effect Estimators.** We compare to a broad suite of baselines. This includes double machine learning (DML) [16, 7, 29], doubly robust learner (DR-Learner) [57, 52], as well as the T-, S-, X-, and domain adaptation learner (DA-Learner), all part of the `EconML` package [11]. Moreover, we include deep-learning–based methods such as TarNet [104], DragonNet [106], and RA-Net [20], implemented via the `CATENets` library [19]. Finally, we compare to inverse propensity weighting (IPW) [97], Bayesian regression trees (BART) [40, 15], and generalized random forests (GRF) [7]. All the baselines, except for IPW, provide both CATE and ATE estimates.

*Importantly, we tune most of the baselines with cross-validation via grid search. The set of hyperparameter, along with the results with default hyperparameters are all detailed in Appendix D.5.*

**Benchmarks with Ground-Truth Effects.** A handful of benchmarks provide ground-truth causal effects, allowing us to directly measure estimation errors. Given a dataset of $N$ units with covariates

---

[2]We use 337 OpenML tables [12], checked to avoid leakage, totaling over $10^9$ feature values.

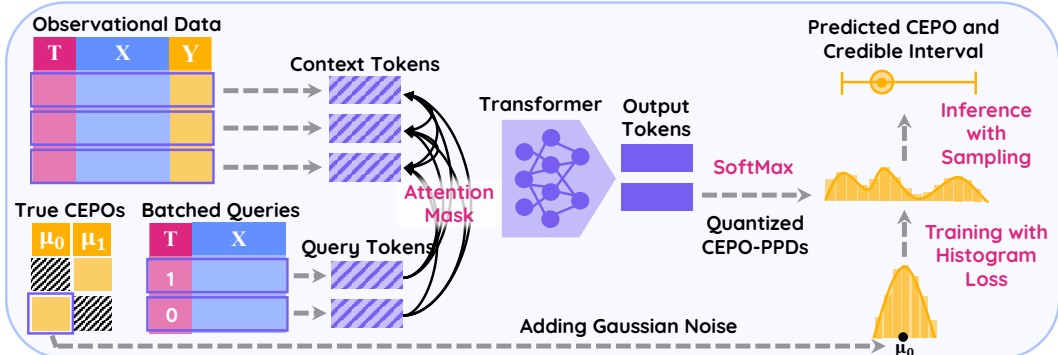

Figure 5: **Architecture, Training, and Inference Details.** *(Left)*: An observational data, and a batch of queries along with their true CEPO values are sampled from the prior. Each observational row forms a context token, while query tokens consist of only the treatment and covariates. *(Middle)*: The context and query tokens are fed into a transformer encoder with an asymmetric attention masking, where both context and query tokens attend only to the context tokens. *(Bottom-Right)*: The output tokens are projected into a 1024-dimensional logit vector and softmaxed to form a discretized CEPO-PPD. Then, the true CEPO value corresponding to each output token is smoothed by adding narrow-width Gaussian, and training is done by minimizing the cross-entropy (histogram) loss. *(Top-Right)*: At inference time, the CEPO-PPD mean is used as the point estimate.

Table 1: **CATE & ATE results.** Columns correspond to benchmark suites: IHDP, ACIC 2016, Lalonde CPS/PSID. *(left half)* mean PEHE and the average rank when pooling all tasks. *(right half)* mean ATE relative error and its average across all tasks. Lalonde PEHE is in thousands. The **best** and **second best** columns are highlighted. Cells with "—" indicate that the method is not applicable.

| Method | Mean PEHE ± Standard Error (↓ better) | | | | | Mean ATE Relative Error ± Standard Error (↓ better) | | | | |
|---|---|---|---|---|---|---|---|---|---|---|
| | IHDP | ACIC 2016 | Lalonde CPS ($\times 10^3$) | Lalonde PSID ($\times 10^3$) | Avg. Rank | IHDP | ACIC 2016 | Lalonde CPS | Lalonde PSID | Avg. Rank |
| **CausalPFN** | 0.58±0.07 | 0.92±0.11 | 8.96±0.02 | 14.40±0.20 | 2.30±0.10 | 0.20±0.04 | 0.05±0.01 | 0.13±0.01 | 0.22±0.02 | 4.45±0.19 |
| T-Learner | 1.73±0.30 | 0.76±0.07 | 9.22±0.04 | 15.16±0.46 | 3.57±0.16 | 0.21±0.04 | 0.03±0.01 | 0.24±0.02 | 0.16±0.03 | 4.31±0.18 |
| DA-Learner | 2.07±0.36 | 0.72±0.08 | 9.39±0.06 | 14.55±0.24 | 3.60±0.16 | 0.23±0.04 | 0.03±0.01 | 0.27±0.02 | 0.20±0.03 | 4.83±0.19 |
| DragonNet | 2.16±0.25 | 2.11±0.19 | 10.93±0.15 | 16.45±0.29 | 5.99±0.18 | 0.20±0.04 | 0.06±0.02 | 0.55±0.03 | 0.47±0.03 | 6.26±0.17 |
| IPW | — | — | — | — | — | 0.24±0.04 | 0.21±0.05 | 0.17±0.01 | 0.10±0.01 | 4.41±0.21 |
| RA-Net | 2.35±0.19 | 2.35±0.25 | 11.74±0.09 | 18.33±0.43 | 7.15±0.16 | 0.20±0.04 | 0.07±0.03 | 0.74±0.02 | 0.50±0.04 | 6.78±0.17 |
| X-Learner | 3.31±0.51 | 0.60±0.08 | 12.15±0.15 | 20.28±0.49 | 7.46±0.19 | 0.16±0.04 | 0.03±0.01 | 0.84±0.03 | 0.72±0.03 | 7.31±0.19 |
| TarNet | 1.82±0.14 | 2.20±0.21 | 12.88±0.02 | 19.19±0.18 | 8.38±0.14 | 0.20±0.04 | 0.05±0.02 | 1.00±0.00 | 0.78±0.01 | 8.83±0.15 |
| S-Learner | 2.57±0.41 | 0.85±0.13 | 12.66±0.05 | 21.80±0.18 | 8.43±0.18 | 0.20±0.04 | 0.03±0.01 | 0.97±0.01 | 0.90±0.02 | 8.85±0.18 |
| BART | 2.50±0.39 | 0.68±0.11 | 12.81±0.05 | 21.36±0.16 | 8.55±0.16 | 0.44±0.09 | 0.04±0.01 | 0.99±0.01 | 0.86±0.01 | 8.99±0.18 |
| GRF | 3.67±0.61 | 1.32±0.30 | 12.33±0.06 | 22.91±0.17 | 8.82±0.18 | 0.18±0.03 | 0.07±0.02 | 0.82±0.02 | 0.85±0.02 | 8.02±0.18 |
| Forest DML | 4.53±0.73 | 1.48±0.31 | 12.95±0.04 | 22.99±0.15 | 9.83±0.17 | 0.08±0.01 | 0.05±0.01 | 1.03±0.01 | 1.05±0.01 | 9.60±0.21 |
| Forest DR Learner | 4.02±0.67 | 1.34±0.29 | 15.98±0.68 | 22.78±0.54 | 10.00±0.17 | 0.17±0.03 | 0.04±0.02 | 1.20±0.23 | 3.64±2.78 | 8.38±0.18 |

and ground-truth CATE values $\{(\mathbf{x}^{(n)}, \tau(\mathbf{x}^{(n)}))\}_{n=1}^{N}$, and a ground-truth ATE $\lambda$, we evaluate models using the relative ATE error and the precision in estimation of heterogeneous effects (PEHE) [40]:

$$\text{RelativeError}(\hat{\lambda}) = \frac{|\hat{\lambda} - \lambda|}{|\lambda|}, \qquad \text{PEHE}(\hat{\tau}) = \sqrt{\frac{1}{N}\sum_{n=1}^{N}\big(\tau(\mathbf{x}^{(n)}) - \hat{\tau}(\mathbf{x}^{(n)})\big)^2}. \qquad (11)$$

Here, $\hat{\tau}$ and $\hat{\lambda}$ denote the estimated CATE and ATE, respectively. Table 1 compares CausalPFN to all baselines on four standard set of datasets: 100 realizations of IHDP [94, 40], 10 realizations of ACIC 2016 [27], and the Lalonde CPS and Lalonde PSID cohorts [58] with their causal effects provided by `RealCause` (each with 100 realizations) [81]. Our model demonstrates superior performance on both CATE and ATE tasks, remaining within the top models across most benchmarks. To assess the overall performance of each method for CATE estimation, we calculate the average rank of each method across all 310 realizations based on PEHE. For ATEs, we calculate the average rank of each method based on relative errors. CausalPFN outperforms all baselines in terms of average CATE rank, while being competitive for average ATE rank. Notably, our model is trained entirely on simulated data and *never* sees the evaluation data during pre-training. While some baseline estimators in Table 1 perform well on specific datasets, they underperform on others. In contrast, the consistent performance of CausalPFN suggests that amortized approaches can potentially eliminate the manual burden of task-specific estimator design.

**Policy Evaluation on Marketing Randomized Trials.** Ground-truth CATEs are only available for synthetic or semi-synthetic datasets. However, if a randomized controlled trial (RCT) is available, we

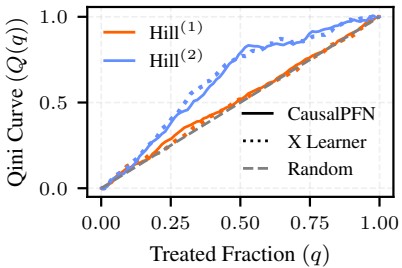

Figure 6: Hill$^{(1)}$ & Hill$^{(2)}$ Qini curves.

Table 2: **Normalized Qini scores** (↑ better). All datasets use 50k stratified subsamples, except Hill$^{(1)}$ and Hill$^{(2)}$, which use the full 64k rows. Columns are normalized to 1.0 for **the best model**.

| Method | Hill$^{(1)}$ | Hill$^{(2)}$ | Criteo | X5 | Lenta | Mega | Avg. |
|---|---|---|---|---|---|---|---|
| **CausalPFN** | 0.992 | 0.968 | 0.859 | 0.922 | **1.000** | 0.970 | **0.952** |
| X Learner | 0.975 | 0.980 | **1.000** | 0.937 | 0.771 | **1.000** | 0.944 |
| S Learner | **1.000** | **1.000** | 0.881 | **1.000** | 0.651 | 0.941 | 0.912 |
| DA Learner | 0.985 | 0.964 | 0.626 | 0.929 | 0.781 | 0.998 | 0.881 |
| T Learner | 0.991 | 0.972 | 0.701 | 0.964 | 0.644 | 0.986 | 0.876 |

can still evaluate the quality of a CATE estimator by assessing the performance of policies derived from it. A common tool for evaluating such policies is the *Qini curve* [93], which plots the cumulative treatment effect when units are ranked in descending order of their predicted CATE.

Formally, let $(y^{(n)}, t^{(n)})_{n=1}^{N}$ denote outcomes and binary treatments from an RCT, and let $\widehat{\tau}_n$ be the corresponding CATE estimates, ordered so that $\widehat{\tau}_1 \geq \cdots \geq \widehat{\tau}_N$. Define

$$\lambda(q) \coloneqq \sum_{n=1}^{\lfloor qN \rfloor} \left( \frac{t^{(n)} y^{(n)}}{r(q)} - \frac{(1-t^{(n)}) y^{(n)}}{1-r(q)} \right), \qquad Q(q) \coloneqq q \cdot \lambda(q)/\lambda(1), \qquad 0 \leq q \leq 1, \qquad (12)$$

where $r(q) = \frac{1}{\lfloor qN \rfloor} \sum_{n}^{\lfloor qN \rfloor} t^{(n)}$ is the empirical treatment rate for the first $q$-quantile of units. Because the data comes from an RCT, $\lambda(q)$ unbiasedly estimates the ATE for the top $q$-quantile of units ranked by predicted CATEs. Plotting $Q(q)$ against the treated fraction $q$ yields the (normalized) Qini curve, and the area under this curve is called the *Qini score*. A random ranking produces a baseline curve as a straight line from $(0,0)$ to $(1,1)$. The higher the Qini curve lies above this line, the better the model prioritizes high-impact units with larger CATE values, leading to greater lift and policy benefit.

We benchmark CausalPFN on five large marketing RCTs from the `scikit-uplift` library [74]. The first dataset, Hillstrom [41], includes 64,000 customers randomly assigned to one of three treatments: no e-mail, an e-mail advertising men's merchandise, or an e-mail advertising women's merchandise. The outcome is whether a website visit occurred within two weeks (binary). We consider two causal tasks: **Hill$^{(1)}$** – Men's-merchandise e-mail (treatment) vs. no e-mail (control), and **Hill$^{(2)}$** – Women's-merchandise e-mail vs. no e-mail. We estimate CATEs using CausalPFN (five-fold honest splitting) and X Learner. Figure 6 shows Qini curves where CausalPFN closely matches X Learner across the targeting range. Notably, Hill$^{(2)}$ shows much greater gains, *suggesting focusing on women's-merchandise ad campaigns, compared to men's, can drive more gains in the number of website visits*. We also evaluate CausalPFN on four larger campaigns—**Lenta**, Retail Hero (**X5**), Megafon (**Mega**), and **Criteo** [61, 95, 78, 122]—each with ~$10^6$ rows. For tractability, we compute Qini scores on stratified 50k subsamples; Table 2 shows CausalPFN achieves the best mean performance. However, when we run it on full tables (see Table 7 of Appendix D.6), we observe a drop in performance, which aligns with known context-length limitations of PFN-style transformers on large tables [109]. Still, the strong subsample results highlight the potential of scaling CausalPFN to longer contexts, which remains an important future direction.

**Uncertainty & Calibration.** Recall from Section 3 that for each unit covariate $\mathbf{x}$, CausalPFN can produce both point estimates and credible intervals for the CATE and CEPOs. We do so by drawing 10,000 samples from the quantized distributions $q_\theta(\cdot \mid \mathbf{x}, t, \mathcal{D}_{\text{obs}})$ and construct credible intervals at any desired significance level $\alpha$. Here, we evaluate these intervals, focusing on the model's calibration. We also assess a key assumption from Proposition 1—whether the inference-time DGP $\psi^\star$ lies within the support of the prior $\pi$, and how the model behaves when this assumption is violated.

We define families of synthetic DGPs to simulate both in-distribution and out-of-distribution (OOD) scenarios. Each DGP samples covariates $\mathbf{x}$ from a uniform distribution, defines a treatment logit function $f$ and CEPO functions $\mu_t$ for $t \in \{0, 1\}$, assigns treatment via $T \sim \text{Bernoulli}(\text{Sigmoid}(f(\mathbf{x})))$, and generates potential outcomes as $y_t = \mu_t(\mathbf{x}) + \epsilon_t$, where $\epsilon_t$ is drawn from a standard Uniform, Gaussian, or Laplace. We consider two DGP families; **Sinusoidal**, where $f$ and $\mu_t$ are functions with sinusoidal components, and **Polynomial**, where the functions $f$ and $\mu_t$ are polynomials of varying degree (see Appendix D.7 for detailed configurations). CausalPFN is trained either on the same family it is tested on, or on a different one (OOD).

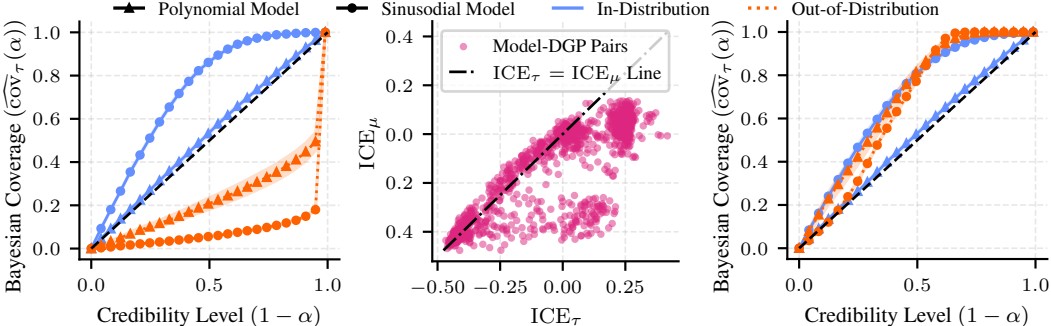

Figure 7: **Calibration.** *(Left)*: CATE coverage vs. nominal credibility. In-distribution DGPs (blue) lie on or above the diagonal (calibrated/conservative), while OOD DGPs (orange) fall below it (overconfident). *(Middle)*: Across model–DGP pairs, CATE ICE (x-axis) exceeds regression ICE (y-axis). *(Right)*: Temperature scaling based on regression ICE ensures the model is either calibrated or conservative for both in- and out-of-distribution DGPs.

For a unit with covariates $\mathbf{x}$ and significance level $\alpha$, we say the true CATE is *covered* if $\tau(\mathbf{x})$ lies within the predicted $100(1-\alpha)\%$ interval obtained using samples from $q_\theta$. Plotting Bayesian coverage against nominal levels of $\alpha$ yields the CATE calibration curve. As shown in Figure 7 (left), CausalPFN is reliably calibrated under in-distribution settings but becomes severely overconfident when evaluated on OOD DGPs ($\psi^\star \not\sim \pi$). This aligns with prior observations that neural models often exhibit pathological overconfidence under distribution shift [35, 86].

To correct this, we apply a temperature parameter $\theta_T$ to the SoftMax that outputs the quantized CEPO-PPD from the logits of the model. We aim to tune $\theta_T$ to minimize the calibration error. However, direct CATE calibration is impossible because $\tau(\mathbf{x})$ is never observed at test-time. Instead, we introduce the *regression calibration* based on observational data: an observed triple $(t, \mathbf{x}, y)$ is covered by the predicted credible interval when $y$ lies inside the model's predicted interval for the CEPO-PPD $\mu_t(\mathbf{x}\ ;\ \psi^\star)$. With that in mind, we let $\widehat{\mathrm{cov}}_\mu(\alpha)$ and $\widehat{\mathrm{cov}}_\tau(\alpha)$ denote the Bayesian coverage at level $\alpha$ for the regression and CATE calibration curves, respectively, and define

$$\mathrm{ICE}_\mu := \int_0^1 (\widehat{\mathrm{cov}}_\mu(\alpha) - \alpha)\ \mathrm{d}\alpha, \quad \text{and} \quad \mathrm{ICE}_\tau := \int_0^1 (\widehat{\mathrm{cov}}_\tau(\alpha) - \alpha)\ \mathrm{d}\alpha, \quad (13)$$

as the integrated coverage error (ICE) for regression and CATE (negative values = overconfidence).

Note that we do not expect $\widehat{\mathrm{cov}}_\mu$ to be calibrated: regression intervals combine epistemic uncertainty of the CEPO with the irreducible (aleatoric) noise in $Y$, so $\mathrm{ICE}_\mu$ is biased. Still, it holds a useful signal. Across all model–DGP pairs in Figure 7 (middle), we consistently observe $\mathrm{ICE}_\mu \le \mathrm{ICE}_\tau$: the regression curve sits at or below the CATE curve. While $\mathrm{ICE}_\tau$ is inaccessible without having the true CATE, $\mathrm{ICE}_\mu$ is computable from observational data. Consequently, temperature-scaling the logits to lift $\widehat{\mathrm{cov}}_\mu$ to the diagonal also calibrates the CATE intervals or makes them conservative. We thus tune $\theta_T$ by grid search to drive $\mathrm{ICE}_\mu$ to zero using a 5-fold calibration on the observational data. The calibrated curves in Figure 7 (right) confirm that, after temperature scaling, CausalPFN's overconfidence on the OOD test-sets disappears. Additional synthetic train-/test-DGP pairs and real-world data experiments appear in Appendix D.7.

**Comparison to TabPFN.** We also compare against the latest version of TabPFN [43], plugging its regression output as a proxy for CEPO. As Table 3 shows, TabPFN is surprisingly competitive without any causal tuning, yet CausalPFN outperforms it on every benchmark except ACIC 2016. To isolate the benefit of training on a causal prior, compared to the predictive *non-identifiable* prior in TabPFN, we fine-tune it on our prior for 48 hours on an H100 GPU. This causal fine-tuning boosts the performance and confirms the added value of identifiable priors for causal effect estimation.

## 6 Related Work

**Single–Dataset Estimators.** Common methods for causal effect estimation are trained and applied on a single dataset. Representative examples include the X-, S-, DR-, and RA-Learners, as well as IPW and DML [11]. Alongside these approaches, several neural variants such as TARNet [104], DragonNet [106], CEVAE [68], and NCMs [114, 115] have been proposed; however, all of them still require per-dataset training and do not amortize across various datasets.

Table 3: **TabPFN Comparison.** PEHE *(left half)* alongside ATE relative error *(right half)*. TabPFN$^\star$ is the latest TabPFN model [43] tuned with our prior. Best numbers are highlighted.

| Method | PEHE ± Standard Error (↓ better) | | | | ATE Relative Error ± Standard Error (↓ better) | | | |
|---|---|---|---|---|---|---|---|---|
| | IHDP | ACIC 2016 | Lalonde CPS ($\times 10^3$) | Lalonde PSID ($\times 10^3$) | IHDP | ACIC 2016 | Lalonde CPS | Lalonde PSID |
| **CausalPFN (Ours)** | **0.58±0.07** | 0.92±0.11 | **8.96±0.02** | **14.40±0.20** | **0.20±0.04** | 0.05±0.01 | **0.13±0.01** | **0.22±0.02** |
| **TabPFN$^\star$ (Ours)** | 0.90±0.16 | **0.47±0.05** | 8.97±0.06 | 14.90±0.95 | 0.21±0.04 | **0.03±0.01** | 0.17±0.02 | 0.22±0.08 |
| TabPFN | 0.95±0.20 | 0.54±0.08 | 9.45±0.19 | 18.7±0.83 | 0.21±0.04 | **0.03±0.01** | 0.32±0.05 | 0.60±0.07 |

**Amortized Causal Inference.** Amortized methods train a *single* network that maps observational data to causal quantities across *multiple* DGPs. Existing approaches fall into two groups: (i) methods that first recover a causal graph from observational data and then compute interventions on that graph [102, 72], following ideas from causal discovery [88, 121, 53, 67, 51, 50]; and (ii) methods that infer causal effects end-to-end [83, 120, 14]. Amortization has also been explored in decision-making, where the aim is to learn policies that generalize across environments or tasks [60, 59]. While closely related, none of these methods provides a ready-to-use estimator that consistently surpasses specialized single-dataset estimators on standard benchmarks. In contrast, our method is trained once and produces causal effects without any access to or adaptation on the test-time DGPs. Through large-scale training, CausalPFN delivers out-of-the-box performance that exceeds specialized single-dataset estimators. Recently, concurrent work by Robertson et al. [96] also applies PFNs to causal effect estimation but lacks a procedure to guarantee the identifiability of the prior data; additionally, we observe relatively poor empirical performance compared to CausalPFN. For further discussion and comparison with this method, refer to Appendix E.

**Scaling In-Context Transformers.** In-context learning with transformers has shown impressive results across a range of domains [13, 116, 18, 25, 110]. Although the underlying mechanisms responsible for this success remain an active area of research [1, 23, 85, 111, 63, 117, 112, 8, 90], increasing model size and training data have consistently and undoubtedly led to stronger performance. This success has recently extended to tabular prediction with models such as TabPFN [42, 43], TabDPT [69], and TabICL [92], which are trained on broad prior distributions and perform well on real-world data without fine-tuning. CausalPFN complements these works, demonstrating that—with sufficient scale and training—in-context learning can also be effectively adapted to causal inference.

## 7 Conclusions, Limitations, and Future Work

In this paper, we introduced a practical paradigm for amortized causal effect estimation that combines Bayesian causal inference with large-scale tabular training. Despite learning solely from simulated data, CausalPFN matches, and often outperforms, specialized causal estimators across diverse real-world domains. Through amortization, we significantly reduce the burden of estimator selection at inference time, and to foster adoption, we have open-sourced the code and presets.

That said, several important limitations remain: *(i)* Our approach fundamentally assumes strong ignorability, which is an untestable assumption in practice. Without this condition, CausalPFN has no guarantees of validity. Domain expertise still remains essential to determine whether this method is appropriate or whether alternative approaches should be used. *(ii)* Our theoretical guarantees rely on idealistic assumptions: a well-specified prior and asymptotically large datasets. We lack finite-sample theory characterizing the estimator's behavior in practical settings. Investigating robustness to prior misspecification and developing finite-sample guarantees remain open problems. Recent work on theory of valid adjustment sets [17] may offer promising directions for addressing these challenges. *(iii)* Performance degradation is evident on the largest marketing tables (Table 7), reflective of the known size-scalability trade-off inherent to PFN-style models [43]. *(iv)* While CausalPFN already supports multi-arm discrete treatments with a finite set $\mathcal{T}$, we have only implemented it for the binary $\mathcal{T}$. Additionally, extending to the continuous treatment setting where $\mathcal{T}$ is not finite remains fully unexplored. *(v)* Finally, our entire implementation relies on the strong ignorability or backdoor assumption. Extending our framework to richer domain-informed priors like instrumental variables can broaden the framework's reach, although designing scalable priors for such cases is non-trivial.

## Acknowledgements

We would like to thank Mouloud Belbahri for his suggestions regarding the uplift modelling experiments. RGK gratefully acknowledges support from the Canada Research Chairs Program (CRC-2022-00049) and the Canada CIFAR AI Chairs Program, This research was funded in part by a NFRF Special Call Award (NFRFR-2022-00526) and NSERC Discovery Grant (RGPIN-2022-04546). Resources used in preparing this research were provided, in part, by the Province of Ontario, the Government of Canada through CIFAR, and companies sponsoring the Vector Institute.

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

# Appendix Contents

# A Notation, Definitions, and Assumptions

**Sample Space.** Let $\mathcal{B}$ denote the Borel $\sigma$-algebra on $\mathbb{R}$. Let $Z = (\mathbf{X}, T, Y)$ collect the observed variables, taking values in a standard Borel space $(\mathcal{Z}, \mathcal{B}_{\mathcal{Z}})$. In particular, $\mathbf{X} \in \mathcal{X}$, $T \in \mathcal{T}$ where $\mathcal{T}$ is finite, and $Y \in \mathbb{R}$. To reason about counterfactuals, define the augmented variable $\tilde{Z} = (\mathbf{X}, T, \{Y_t\}_{t \in \mathcal{T}}, Y)$ on a standard Borel space $(\tilde{\mathcal{Z}}, \mathcal{B}_{\tilde{\mathcal{Z}}})$.

**Data–Generating Parameters.** Let $(\Psi, \mathcal{B}_{\Psi})$ be a standard Borel parameter space. For $\psi \in \Psi$, a data–generating process (DGP) is a probability measure $P^{\psi}$ on $(\tilde{\mathcal{Z}}, \mathcal{B}_{\tilde{\mathcal{Z}}})$, which induces the observational marginal $P^{\psi}_{\text{obs}}$ on $(\mathcal{Z}, \mathcal{B}_{\mathcal{Z}})$. We use $\psi$ to denote both the random parameter (when distributed according to a prior) and its realized value, when clear from context.

We impose a mild regularity condition to ensure measurability of parameter–to–law maps. Let $\mathcal{P}(\tilde{\mathcal{Z}})$ and $\mathcal{P}(\mathcal{Z})$ denote the spaces of probability measures on $\tilde{\mathcal{Z}}$ and $\mathcal{Z}$, respectively, endowed with the Borel $\sigma$-algebras generated by the weak topologies.

**Assumption 2** (Measurability). The map $\psi \mapsto P^{\psi} \in \mathcal{P}(\tilde{\mathcal{Z}})$ is measurable, in the sense that $\psi \mapsto P^{\psi}(B)$ is $\mathcal{B}_{\Psi}$-measurable for each $B \in \mathcal{B}_{\tilde{\mathcal{Z}}}$. Similarly, $\psi \mapsto P^{\psi}_{\text{obs}} \in \mathcal{P}(\mathcal{Z})$ is $\mathcal{B}_{\Psi}$-measurable and its image set $\{P^{\psi}_{\text{obs}} : \psi \in \Psi\}$ is a Borel subset of $\mathcal{P}(\mathcal{Z})$.

**Prior and Posterior Distributions.** Let $\pi$ be a prior on $(\Psi, \mathcal{B}_{\Psi})$. Define the joint law $P^{\pi}$ of $\left( (\tilde{Z}_i)_{i \geq 1}, \psi \right)$ by first sampling $\psi \sim \pi$ and then, conditional on $\psi$, sampling $(\tilde{Z}_i)_{i \geq 1}$ i.i.d. from $P^{\psi}$. We use $P^{\pi}_{\mathbf{X}}$ to denote its marginal distribution on $\mathbf{X}$.

Let $\mathcal{D}^n_{\text{obs}} := (Z_1, Z_2, \ldots, Z_n)$ denote the first $n$ observed variables (the $\mathcal{Z}$-marginals of the corresponding $\tilde{Z}_i$). We write $\pi(\cdot \mid \mathcal{D}^n_{\text{obs}})$ for the posterior on $\Psi$ induced by $P^{\pi}$.

**Parametric CEPOs and CEPO Posterior Predictive.** For each $t \in \mathcal{T}$ and $\pi$-almost every $\psi$, regular conditional distributions for $(Y_t \mid \mathbf{X})$ exist because all relevant spaces are standard Borel. Thus, there is a Borel version of the conditional expectation $\mathbf{x} \mapsto \mathbb{E}^{P^{\psi}}[Y_t \mid \mathbf{X} = \mathbf{x}]$. We fix a version $\mu_t(\cdot \ ; \ \psi)$ that is jointly measurable in $(\mathbf{X}, \psi)$ and call it the conditional expected potential outcome (CEPO):

$$\mu_t(\mathbf{x} \ ; \ \psi) := \mathbb{E}^{P^{\psi}}[Y_t \mid \mathbf{X} = \mathbf{x}], \qquad \text{for } P^{\pi}_{\mathbf{X}}\text{-almost every } \mathbf{x}. \tag{14}$$

**Assumption 3** (Integrability). For every $t \in \mathcal{T}$ and $P^{\pi}_{\mathbf{X}}$-almost every $\mathbf{x}$:

$$\mathbb{E}^{\pi}\big[ |\mu_t(\mathbf{x} \ ; \ \psi)| \big] < \infty. \tag{15}$$

For any query $(t, \mathbf{x})$ and dataset $\mathcal{D}^n_{\text{obs}}$, the CEPO posterior predictive distribution (CEPO-PPD), a probability measure on $\mathbb{R}$, is the pushforward of the posterior $\pi(\psi \mid \mathcal{D}^n_{\text{obs}})$ through $\psi \mapsto \mu_t(\mathbf{x}; \psi)$:

$$\pi^{\mu_t}(B \mid \mathbf{x}, \mathcal{D}^n_{\text{obs}}) := \int_{\Psi} \mathbb{I}(\mu_t(\mathbf{x} \ ; \ \psi) \in B) \pi(\mathrm{d}\psi \mid \mathcal{D}^n_{\text{obs}}), \qquad B \in \mathcal{B}. \tag{16}$$

**Model.** Given a query $(t, \mathbf{x})$ and context $\mathcal{D}^n_{\text{obs}}$, a model with parameters $\theta$ yields a predictive distribution $q_{\theta}(\cdot \mid \mathbf{x}, t, \mathcal{D}^n_{\text{obs}})$ on $\mathbb{R}$ for the CEPO values.

**Observational Quotient Space.** Standard consistency results such as Doob's consistency theorem [79] are concerned with the parameters of the *observational* distribution. To leverage such results, we characterize the set of DGPs with the same observational distributions as follows:

**Definition 4** (Observational Quotient Space). Let $\Phi := \Psi/\sim$ be the set of equivalence classes under: $\psi_1 \sim \psi_2$ iff $P^{\psi_1}_{\text{obs}} = P^{\psi_2}_{\text{obs}}$. Let $[\cdot] : \Psi \to \Phi$ be the quotient map and equip $\Phi$ with the quotient $\sigma$-algebra $\mathcal{B}_{\Phi} := \{A \subseteq \Phi : [\cdot]^{-1}(A) \in \mathcal{B}_{\Psi}\}$. We call $(\Phi, \mathcal{B}_{\Phi})$ the observational quotient space.

We write $\phi = [\psi]$ to denote the equivalence class corresponding to the parameter $\psi$ and may interchangeably use $P^{[\psi]}$, $P^{\phi}$, or $P^{\psi}_{\text{obs}}$ to denote the corresponding observational distribution. Note $(\Phi, \mathcal{B}_{\Phi})$ is also standard Borel. Indeed, identify $\Phi$ with the image $R := \{P^{\psi}_{\text{obs}} : \psi \in \Psi\} \subseteq \mathcal{P}(\mathcal{Z})$ via the measurable bijection $[\psi] \mapsto P^{\psi}_{\text{obs}}$. By Assumption 2, $R$ is a Borel subset of the standard Borel space $\mathcal{P}(\mathcal{Z})$ (the space of probability measures on $\mathcal{Z}$ with the weak topology is standard Borel whenever $(\mathcal{Z}, \mathcal{B}_{\mathcal{Z}})$ is), hence $R$ and thus $\Phi$ are standard Borel [108].

**Identifiability.** Finally, we re-state the definition of identifiability from Section 2 more formally. This will be a necessary condition to prove our results in the next sections.

**Definition 1** (CEPO-Identifiability). We call a prior $\pi$ on $(\Psi, \mathcal{B}_\Psi)$ CEPO-identifiable, if for each value of $t \in \mathcal{T}$, there exists a map $f_t : \mathcal{X} \times \Phi \to \mathbb{R}$, such that for $\pi$-almost all parameters $\psi$ and $P_{\mathbf{X}}^\pi$-almost all values of $\mathbf{x}$, the CEPO value $\mu_t(\mathbf{x} \; ; \; \psi) = f_t(\mathbf{x}, [\psi])$.

Note that the above definition is compatible with the standard identifiability in the literature [89], since there is a bijection between each equivalence class $[\psi]$ and the observational distribution $P_{\text{obs}}^\psi$.

# B  Consistency Result

## B.1  Re-Statement of Proposition 1

**Proposition 1** (Formal). *Under Assumptions 2 and 3, there exist sets $\mathcal{X}_0 \subseteq \mathcal{X}$ and $\Psi^\star \subseteq \Psi$ with $P_{\mathbf{X}}^\pi(\mathcal{X}_0) = 1$ and $\pi(\Psi^\star) = 1$, such that for all $t \in \mathcal{T}, \mathbf{x} \in \mathcal{X}_0$, and $\psi^\star \in \Psi^\star$, if $Z_1, Z_2, \ldots \sim P_{\text{obs}}^{\psi^\star}$ i.i.d., then*

$$\lim_{n \to \infty} \mathbb{E}_{\mu \sim \pi^{\mu_t}(\cdot \mid \mathbf{x}, \mathcal{D}_{\text{obs}}^n)}[\mu] = \mu_t(\mathbf{x} \; ; \; \psi^\star) \quad P_{\text{obs}}^{\psi^\star}\text{-a.s.}, \tag{17}$$

*if and only if the prior $\pi$ is CEPO-identifiable.*

## B.2  Preliminaries for the Proof of Proposition 1

Here, we introduce some concepts to simplify the statement of the proof. We start by presenting a corollary of Doob's consistency theorem without proof (Corollary 2.3 of Miller [79]), which we will heavily leverage for the proof of Proposition 1. The result is re-stated to match our parallel notation:

**Theorem 2** (Corollary of Doob's Consistency Theorem). *Suppose $(\mathcal{Z}, \mathcal{B}_\mathcal{Z})$ and $(\Phi, \mathcal{B}_\Phi)$ are two standard Borel spaces. Let $\nu$ be a probability measure on $(\Phi, \mathcal{B}_\Phi)$. For each $\phi \in \Phi$, let $P^\phi$ be a probability measure on $(\mathcal{Z}, \mathcal{B}_\mathcal{Z})$. Consider a measurable map $g : \Phi \to \mathbb{R}$ and assume:*

- *(i)* **Measurability.** *$\phi \mapsto P^\phi(B)$ is measurable for every $B \in \mathcal{B}_\mathcal{Z}$.*

- *(ii)* **No Redundancy.** *$\phi \neq \phi' \implies P^\phi \neq P^{\phi'}$.*

- *(iii)* **Integrability.** *$\mathbb{E}^\nu[|g(\phi)|] < \infty$.*

*Moreover, define the extended joint probability measure $\nu_{tot}$ on $((Z_1, Z_2, \ldots), \phi)$ by first drawing $\phi \sim \nu$, and then, conditioned on $\phi$, sampling $Z_1, Z_2, \ldots$ i.i.d. from $P^\phi$. Then, there exists $\Phi_0 \subseteq \Phi$ with $\nu(\Phi_0) = 1$, such that for any $\phi_0 \in \Phi_0$ and $Z_1, Z_2, \ldots \sim P^{\phi_0}$ i.i.d., we have*

$$\lim_{n \to \infty} \mathbb{E}^{\nu_{tot}}[g(\phi) \mid Z_1, \ldots, Z_n] = g(\phi_0) \quad P^{\phi_0}\text{-a.s.} \tag{18}$$

**The Joint Measure $\Pi$.** For technical convenience, we define a joint measure $\Pi$ on variables $\psi, (\tilde{Z}_i)_{i \geq 1}, [\psi]$, and $(Z_i)_{i \geq 1}$, as the pushforward measure of $P^\pi$ by the following map:

$$\left( \psi, (\tilde{Z}_i)_{i \geq 1} \right) \mapsto \left( \psi, (\tilde{Z}_i)_{i \geq 1}, [\psi], (Z_i)_{i \geq 1} \right). \tag{19}$$

In particular, we have the following equalities:

$$\Pi((Z_i)_{i \geq 1} \mid \psi, [\psi]) = P_{\text{obs}}^\psi((Z_i)_{i \geq 1}) = P^{[\psi]}((Z_i)_{i \geq 1}) = \Pi((Z_i)_{i \geq 1} \mid [\psi]), \tag{20}$$

which results in the conditional independence

$$(Z_i)_{i \geq 1} \perp\!\!\!\perp_\Pi \psi \mid [\psi]. \tag{21}$$

Since all spaces involved are standard Borel, regular conditional distributions exist; hence the above conditional laws are well defined [91].

*(Notation Remark)* We remove the superscript $\Pi$ in expectations and simply write $\mathbb{E}$ when we take expectations w.r.t. $\Pi$. Also, we reuse the symbol $\Pi$ for the joint measure and for any of its marginals or conditionals; the intended meaning will be clear from context.

## B.3   Proof of Proposition 1

For any given $t$ and $\mathbf{x}$, define the expected CEPOs under observational equivalence class $\phi \in \Phi$ as[3]

$$g_t(\mathbf{x} ; \phi) := \mathbb{E}[\mu_t(\mathbf{x} ; \psi) \mid \phi]. \tag{22}$$

We can use Theorem 2 to establish a consistency result connecting the CEPO-PPDs and the functions $g_t$ defined in (22):

**Lemma 3.** *Under Assumptions 2 and 3, there exist sets $\mathcal{X}_0 \subseteq \mathcal{X}$ and $\Psi_0 \subseteq \Psi$ with $P_{\mathbf{X}}^\pi(\mathcal{X}_0) = 1$ and $\pi(\Psi_0) = 1$, such that for all $t \in \mathcal{T}, \mathbf{x} \in \mathcal{X}_0$, and $\psi_0 \in \Psi_0$, if $Z_1, Z_2, \ldots \sim P_{\mathrm{obs}}^{\psi_0}$ i.i.d., then*

$$\lim_{n \to \infty} \mathbb{E}_{\mu \sim \pi^{\mu_t}(\cdot \mid \mathbf{x}, \mathcal{D}_{\mathrm{obs}}^n)}[\mu] = g_t(\mathbf{x} ; [\psi_0]) \quad P_{\mathrm{obs}}^{\psi_0}\text{-a.s.} \tag{23}$$

*where $\mu$ denotes the identity map on $\mathbb{R}$.*

*Proof.* From Assumption 3, a subset $\mathcal{X}_0 \subseteq \mathcal{X}$ exists with $P_{\mathbf{X}}^\pi(\mathcal{X}_0) = 1$, such that for all $t \in \mathcal{T}$ and $\mathbf{x} \in \mathcal{X}_0$, we have $\mathbb{E}^\pi[|\mu_t(\mathbf{x} ; \psi)|] < \infty$. Fix a value of $\mathbf{x}_0 \in \mathcal{X}_0$ and $t_0 \in \mathcal{T}$. A similar integrability statement can be made for $g_{t_0}(\mathbf{x}_0 ; \phi)$:

$$\begin{aligned}
\mathbb{E}[|g_{t_0}(\mathbf{x}_0 ; \phi)|] &= \mathbb{E}[|\mathbb{E}[\mu_{t_0}(\mathbf{x}_0 ; \psi) \mid \phi]|] &&\text{from (22)} \\
&\leq \mathbb{E}[\mathbb{E}[|\mu_{t_0}(\mathbf{x}_0 ; \psi)| \mid \phi]] &&\text{(Jensen's inequality)} \\
&= \mathbb{E}[|\mu_{t_0}(\mathbf{x}_0 ; \psi)|] < \infty. &&\text{(total expectation)} \tag{24}
\end{aligned}$$

Now, we use Theorem 2 to obtain the desired results for the function $g_{t_0}(\mathbf{x}_0 ; \phi)$ by plugging in $(\mathcal{Z}, \mathcal{B}_\mathcal{Z})$ directly from our notation and $(\Phi, \mathcal{B}_\Phi)$ from Definition 4. Moreover, we replace $\nu$ and $\nu_{\mathrm{tot}}$ by the marginals of $\Pi$ on the random variables $\phi$ and $((Z_i)_{i \geq 1}, \phi)$, respectively. Finally, it is easy to see that all the required assumptions hold:

- *(i)* **Measurability.** Follows from the measurability in Assumption 2.

- *(ii)* **No Redundancy.** Follows from the definition of the quotient space in Definition 4.

- *(iii)* **Integrability.** Follows from (24).

As a result of Theorem 2, there exists a set $\Phi_0 \subseteq \Phi$ with $\Pi(\Phi_0) = 1$, such that for any $\phi_0 \in \Phi_0$ and $Z_1, Z_2, \ldots \sim P^{\phi_0}$ i.i.d., we have

$$\lim_{n \to \infty} \mathbb{E}[g_{t_0}(\mathbf{x}_0 ; \phi) \mid \mathcal{D}_{\mathrm{obs}}^n] = g_{t_0}(\mathbf{x}_0 ; \phi_0) \quad P^{\phi_0}\text{-a.s.} \tag{25}$$

We can simplify the expectation in the L.H.S. of (25) as follows:

$$\begin{aligned}
\mathbb{E}[g_{t_0}(\mathbf{x}_0 ; \phi) \mid \mathcal{D}_{\mathrm{obs}}^n] &= \mathbb{E}[\mathbb{E}[\mu_{t_0}(\mathbf{x}_0 ; \psi) \mid \phi] \mid \mathcal{D}_{\mathrm{obs}}^n] &&\text{from (22)} \\
&= \mathbb{E}[\mathbb{E}[\mu_{t_0}(\mathbf{x}_0 ; \psi) \mid \mathcal{D}_{\mathrm{obs}}^n, \phi] \mid \mathcal{D}_{\mathrm{obs}}^n] &&\text{from (21)} \\
&= \mathbb{E}[\mu_{t_0}(\mathbf{x}_0 ; \psi) \mid \mathcal{D}_{\mathrm{obs}}^n] &&\text{(tower property)} \\
&\overset{(\star)}{=} \mathbb{E}_{\mu \sim \pi^{\mu_{t_0}}(\cdot \mid \mathbf{x}_0, \mathcal{D}_{\mathrm{obs}}^n)}[\mu], \tag{26}
\end{aligned}$$

where $(\star)$ follows from the fact that CEPO-PPD $\pi^{\mu_{t_0}}$ is the pushforward of the posterior $\Pi(\psi \mid \mathcal{D}_{\mathrm{obs}}^n) = \pi(\psi \mid \mathcal{D}_{\mathrm{obs}}^n)$ under the map $\psi \mapsto \mu_{t_0}(\mathbf{x}_0 ; \psi)$.

We then define $\Psi_0$ as the preimage of $\Phi_0$ under the quotient mapping. It is easy to verify that $\pi(\Psi_0) = \Pi(\Psi_0) = \Pi(\Phi_0) = 1$. For any $\psi_0 \in \Psi_0$, set $\phi_0 = [\psi_0]$. Combining (25) and (26), and repeating the entire argument for all $t_0 \in \mathcal{T}$ and $\mathbf{x}_0 \in \mathcal{X}_0$ concludes the proof. $\qquad\square$

Lemma 3 establishes a consistency result between the CEPO-PPDs and functions $g_t$ we defined on the quotient space. With the consistency proven in the observational quotient space, all that remains is to connect the R.H.S. of (23) to the original CEPOs. This is where identifiability comes into play. In what follows, fix $t_0 \in \mathcal{T}$ and $\mathbf{x}_0 \in \mathcal{X}_0$:

---

[3]Equivalently, $g_t(\mathbf{x} ; \phi) = \mathbb{E}[\mu_t(\mathbf{x} ; \psi) \mid [\psi] = \phi]$.

**CEPO-Identifiability $\Rightarrow$ Consistency.** Under CEPO-identifiability (Definition 1), there exists $\Psi_1 \subseteq \Psi$ with $\pi(\Psi_1) = 1$, where for all $\psi', \psi'' \in \Psi_1$ that $[\psi'] = [\psi'']$, we have $\mu_{t_0}(\mathbf{x}_0 ; \psi') = \mu_{t_0}(\mathbf{x}_0 ; \psi'')$. Define $\Psi^\star := \Psi_1 \cap \Psi_0$ and note that $\pi(\Psi^\star) = 1$. Consequently, for any $\psi^\star \in \Psi^\star$, we also have $\psi^\star \in \Psi_1$, and

$$g_{t_0}(\mathbf{x}_0 ; [\psi^\star]) \overset{(22)}{=} \mathbb{E}[\mu_{t_0}(\mathbf{x}_0 ; \psi) \mid [\psi] = [\psi^\star]] = \mu_{t_0}(\mathbf{x}_0 ; \psi^\star). \tag{27}$$

Combining (27) with Lemma 3 and repeating the argument for all $t_0 \in \mathcal{T}$ and $\mathbf{x}_0 \in \mathcal{X}_0$ proves the first side of Proposition 1.

**Consistency $\Rightarrow$ CEPO-Identifiability.** When consistency holds, from (17), a set $\Psi^\star \subseteq \Psi$ exists with $\pi(\Psi^\star) = 1$, where for all $\psi^\star \in \Psi^\star$, if $Z_1, Z_2, \ldots \sim P_{\text{obs}}^{\psi^\star}$, then

$$\lim_{n \to \infty} \mathbb{E}_{\mu \sim \pi^{\mu_{t_0}} \left( \cdot \middle| \mathbf{x}_0, \mathcal{D}_{\text{obs}}^n \right)}[\mu] = \mu_{t_0}(\mathbf{x}_0 ; \psi^\star) \quad P_{\text{obs}}^{\psi^\star}\text{-a.s.} \tag{28}$$

Moreover, according to Lemma 3, there exists a set $\Psi_0 \subseteq \Psi$ with $\pi(\Psi_0) = 1$, such that for all $\psi_0 \in \Psi_0$, if $Z_1, Z_2, \ldots \sim P_{\text{obs}}^{\psi_0}$, then

$$\lim_{n \to \infty} \mathbb{E}_{\mu \sim \pi^{\mu_{t_0}} \left( \cdot \middle| \mathbf{x}_0, \mathcal{D}_{\text{obs}}^n \right)}[\mu] = g_{t_0}(\mathbf{x}_0 ; [\psi_0]) \quad P_{\text{obs}}^{\psi_0}\text{-a.s.} \tag{29}$$

Using these two identities, we can define $\Psi_1 := \Psi^\star \cap \Psi_0$, where $\pi(\Psi_1) = 1$, and the following holds for every $\psi_1 \in \Psi_1$:

$$\mu_{t_0}(\mathbf{x}_0 ; \psi_1) = g_{t_0}(\mathbf{x}_0 ; [\psi_1]). \tag{30}$$

Hence, the prior $\pi$ is indeed CEPO-identifiable, as we can use the functional $g$ in place of $f$ in Definition 1. Repeating this process for all $t_0 \in \mathcal{T}$ and $\mathbf{x}_0 \in \mathcal{X}_0$ concludes the proof of Proposition 1.

# C   Validity of the Causal Data-Prior Loss

Here, we show that the causal data-prior loss is equivalent to the expected forward KL divergence between the CEPO-PPDs and the parameterized distribution $q_\theta$. For the theoretical justification, we assume a fixed observation size $n$ and define $\mathcal{D}_{\text{obs}} := \mathcal{D}_{\text{obs}}^n$ with a dropped superscript for simplicity.

**Assumption 4** (Existence of Densities)**.** We assume each CEPO-PPD $\pi^{\mu_t}(\cdot \mid \mathbf{x}, \mathcal{D}_{\text{obs}}^n)$ admits a density w.r.t. Lebesgue measure and use the same symbol for its density. Moreover, we assume $q_\theta(\cdot \mid \mathbf{x}, t, \mathcal{D}_{\text{obs}}^n)$ is a probability measure with full support on $\mathbb{R}$, which admits a density w.r.t. Lebesgue measure. Similar to CEPO-PPDs, we use the same symbol for the measure and its density.

**Definition 5.** Let $P_{\text{obs}}^\pi$ be the marginal distribution of $P^\pi$ on $(Z_i)_{i \geq 1}$. Then, the expected forward-KL divergence between $\pi^{\mu_t}$ and $q_\theta$ is defined as

$$\mathcal{L}_t^{\text{KL}}(\theta) := \mathbb{E}_{\mathcal{D}_{\text{obs}} \cup \{\mathbf{x}\} \sim P_{\text{obs}}^\pi}[\text{D}_{\text{KL}}(\pi^{\mu_t}(\cdot \mid \mathbf{x}, \mathcal{D}_{\text{obs}}) \| q_\theta(\cdot \mid \mathbf{x}, t, \mathcal{D}_{\text{obs}}))], \tag{31}$$

where $\mathcal{D}_{\text{obs}} \cup \{\mathbf{x}\} \sim P_{\text{obs}}^\pi$ refers to first drawing $\psi \sim \pi$, and then sampling $\mathcal{D}_{\text{obs}} = (Z_1, \ldots, Z_n)$ i.i.d. from $P_{\text{obs}}^\psi$ and an independent query point $\mathbf{x} \sim P_{\mathbf{X}}^\psi$.

**Proposition 4.** *Under Assumption 4, the causal data-prior loss from Definition 3 and the expected forward-KL divergence in Definition 5 have the same optima. In other words, for all $t \in \mathcal{T}$,*

$$\arg\min_\theta \mathcal{L}_t^{\text{KL}}(\theta) = \arg\min_\theta \mathcal{L}_t(\theta). \tag{32}$$

*Proof.* Fix a $t \in \mathcal{T}$. We note that

$$\mathcal{L}_t^{\text{KL}}(\theta) = \mathbb{E}_{\mathcal{D}_{\text{obs}} \cup \{\mathbf{x}\} \sim P_{\text{obs}}^\pi}\left[\mathbb{E}_{\mu \sim \pi^{\mu_t}(\cdot \mid \mathbf{x}, \mathcal{D}_{\text{obs}})}\left[\log \frac{\pi^{\mu_t}(\mu \mid \mathbf{x}, \mathcal{D}_{\text{obs}})}{q_\theta(\mu \mid \mathbf{x}, t, \mathcal{D}_{\text{obs}})}\right]\right]. \tag{33}$$

From (16), we know that $\pi^{\mu_t}(\cdot \mid \mathbf{x}, \mathcal{D}_{\text{obs}})$ is the pushforward of the posterior $\pi(\cdot \mid \mathcal{D}_{\text{obs}})$ by the function $\psi \mapsto \mu_t(\mathbf{x} ; \psi)$. Hence, for any measurable function $h : \mathbb{R} \to \mathbb{R}$, we get

$$\mathbb{E}_{\mu \sim \pi^{\mu_t}(\cdot \mid \mathbf{x}, \mathcal{D}_{\text{obs}})}[h(\mu)] = \mathbb{E}_{\psi \sim \pi(\cdot \mid \mathcal{D}_{\text{obs}})}[h(\mu_t(\mathbf{x} ; \psi))]. \tag{34}$$

Setting $h(\mu) = \log \frac{\pi^{\mu_t}(\mu \mid \mathbf{x}, \mathcal{D}_{\mathrm{obs}})}{q_\theta(\mu \mid \mathbf{x}, t, \mathcal{D}_{\mathrm{obs}})}$ in (34) and combining with (33) yields

$$\mathcal{L}_t^{\mathrm{KL}}(\theta) = \mathbb{E}_{\mathcal{D}_{\mathrm{obs}} \cup \{\mathbf{x}\} \sim P_{\mathrm{obs}}^\pi} \left[ \mathbb{E}_{\psi \sim \pi(\cdot \mid \mathcal{D}_{\mathrm{obs}})} \left[ \log \frac{\pi^{\mu_t}(\mu_t(\mathbf{x} \; ; \; \psi) \mid \mathbf{x}, \mathcal{D}_{\mathrm{obs}})}{q_\theta(\mu_t(\mathbf{x} \; ; \; \psi) \mid \mathbf{x}, t, \mathcal{D}_{\mathrm{obs}})} \right] \right] \tag{35}$$

$$= \mathbb{E}_{\mathcal{D}_{\mathrm{obs}} \cup \{\mathbf{x}\} \sim P_{\mathrm{obs}}^\pi, \; \psi \sim \pi(\cdot \mid \mathcal{D}_{\mathrm{obs}})} \left[ \log \frac{\pi^{\mu_t}(\mu_t(\mathbf{x} \; ; \; \psi) \mid \mathbf{x}, \mathcal{D}_{\mathrm{obs}})}{q_\theta(\mu_t(\mathbf{x} \; ; \; \psi) \mid \mathbf{x}, t, \mathcal{D}_{\mathrm{obs}})} \right]. \tag{36}$$

Next, we use the Bayes' rule to derive

$$\underbrace{P_{\mathrm{obs}}^\pi(\mathcal{D}_{\mathrm{obs}})}_{\text{evidence}} \underbrace{\pi(\psi \mid \mathcal{D}_{\mathrm{obs}})}_{\text{posterior}} = \underbrace{\pi(\psi)}_{\text{prior}} \underbrace{P_{\mathrm{obs}}^\psi(\mathcal{D}_{\mathrm{obs}})}_{\text{likelihood}}. \tag{37}$$

Combining (36) and (37), we get

$$\mathcal{L}_t^{\mathrm{KL}}(\theta) = \mathbb{E}_{\psi \sim \pi, \; \mathcal{D}_{\mathrm{obs}} \cup \{\mathbf{x}\} \sim P_{\mathrm{obs}}^\psi} \left[ \log \frac{\pi^{\mu_t}(\mu_t(\mathbf{x} \; ; \; \psi) \mid \mathbf{x}, \mathcal{D}_{\mathrm{obs}})}{q_\theta(\mu_t(\mathbf{x} \; ; \; \psi) \mid \mathbf{x}, t, \mathcal{D}_{\mathrm{obs}})} \right] \tag{38}$$

$$= \mathbb{E}_{\psi \sim \pi, \; \mathcal{D}_{\mathrm{obs}} \cup \{\mathbf{x}\} \sim P_{\mathrm{obs}}^\psi} \left[ -\log q_\theta(\mu_t(\mathbf{x} \; ; \; \psi) \mid \mathbf{x}, t, \mathcal{D}_{\mathrm{obs}}) \right] + \text{constant term in } \theta \tag{39}$$

$$= \mathcal{L}_t(\theta) + \text{constant term in } \theta, \tag{40}$$

which concludes the proof. $\qquad\square$

*(Remark 1)* In general, the forward-KL divergence loss cannot be estimated without estimating the true CEPO-PPD. However, with this identity established, we can justify the use of the equivalent causal data-prior loss which is easily estimable.

*(Remark 2)* The theoretical equivalence is proved only for a fixed treatment $t \in \mathcal{T}$ and a fixed, finite sample size $n \in \{1, 2, \ldots\}$. In practice, the training loss is minimized while *randomizing* both $t$ and the sample size $n$. If the optimizer attains a near-optima of this randomized objective, the approximation $q_\theta(\cdot \mid \mathbf{x}, t, \mathcal{D}_{\mathrm{obs}}) \approx \pi^{\mu_t}(\cdot \mid \mathbf{x}, \mathcal{D}_{\mathrm{obs}})$ can effectively extend to all the treatment values and to almost every sample size we care about in practice.

# D  Experimental Details

## D.1  Prior Generation & Simulating DGPs

As illustrated in Figure 4, our prior generation consists of retrieving or synthesizing a base table, subsampling covariates $\mathbf{X}$ and CEPOs $\mu_0$ and $\mu_1$, synthesizing treatments $T$, potential outcomes $Y_t$, and finally, observed outcomes $Y$. We break down each of the components:

**Data Sources for the Base Tables.** We draw the base tables from two sources: *(i)* real-world tables from OpenML, and *(ii)* fully synthetic data.

  *(i)* We use the OpenML collections used in Grinsztajn et al. [34], AMLB [33], and TabZilla [77], all listed in Ma et al. [69]. To widen coverage, we also add tables from CTR23 [28] and CC18 [12]. All OpenML IDs are in this link.[4] Data leakage is ruled out as none of the tables that share covariates or outcomes with our test sets (Lalonde, IHDP, ACIC, Criteo, Megafon, Hillstrom, Lenta, X5) are included in training. Moreover, the propensities are sampled purely synthetically, following the approach described below.

  *(ii)* For additional diversity, we generate synthetic tables using the random neural networks used to train TabPFN v1, with the same hyperparameters described in Hollmann et al. [42]. Inputs, from a standard Gaussian distribution, are fed into the network, and a subset of the outputs and hidden neurons are selected to construct the tabular data. Some columns are discretized at random to produce categorical and ordinal variables to reflect the structure of real-world tabular domains. While TabPFN v2 [43] is a newer and stronger model, its training data is not publicly available, so we restrict ourselves to the v1 generator to ensure transparent evaluation and leakage control.

---

[4] https://drive.google.com/file/d/1NXib83Lc7jGOPJx554p-I3sxFrcWeF52

**CEPOs with Heterogeneity Control.** Once the base table is given, we randomly select two columns and name them $\mu_{\text{raw},0}$ and $\mu_{\text{raw},1}$. However, in practice, we observe that directly using such columns for CEPOs can result in large variances (*heterogeneity*) for CATEs. We therefore apply a light-weight post-processing inspired by `RealCause` [81].

The post-processing requires a heterogeneity hyperparameter $\gamma$, which we sample uniformly from $[0, 1]$ during prior generation. Then, for $N$ units (rows) extracted from the base table, let $\tau_{\text{raw}}^{(n)} = \mu_{\text{raw},1}^{(n)} - \mu_{\text{raw},0}^{(n)}$ be the CATE for unit $n \in [N]$, and $\lambda_{\text{raw}} = \frac{1}{N} \sum_{n=1}^{N} \tau_{\text{raw}}^{(n)}$ the sample ATE. We draw i.i.d. $\{\alpha^{(n)}\}_{n=1}^{N} \sim \text{Unif}[0, 1]$ and construct the final $\gamma$-augmented CEPOs as

$$\mu_1^{(n)} := \left[\alpha^{(n)} + (1 - \alpha^{(n)})\gamma\right]\mu_{\text{raw},1}^{(n)} + (1 - \gamma)(1 - \alpha^{(n)})(\mu_{\text{raw},0}^{(n)} + \lambda_{\text{raw}}), \tag{41}$$

$$\mu_0^{(n)} := \left[(1 - \alpha^{(n)}) + \alpha^{(n)}\gamma\right]\mu_{\text{raw},0}^{(n)} + (1 - \gamma)\alpha^{(n)}(\mu_{\text{raw},1}^{(n)} - \lambda_{\text{raw}}). \tag{42}$$

A simple algebraic check shows

$$\tau^{(n)} := \mu_1^{(n)} - \mu_0^{(n)} = \gamma\tau_{\text{raw}}^{(n)} + (1 - \gamma)\lambda_{\text{raw}}, \quad \text{Var}[\tau \mid \mathbf{x}] = \gamma^2 \text{Var}[\tau_{\text{raw}} \mid \mathbf{x}]. \tag{43}$$

Hence, while preserving the average treatment effect, $\gamma = 0$ yields a dataset with a zero variance CATE (fully homogeneous), whereas $\gamma = 1$ recovers the original heterogeneity.

**Outcomes.** After constructing the CEPO columns $\mu_0(\mathbf{x})$ and $\mu_1(\mathbf{x})$, we need to turn them into potential outcomes by adding zero-mean noises. To avoid tying the data to a specific parametric noise model, we introduce two additional *nuisance* columns, $\eta_0(\mathbf{x})$ and $\eta_1(\mathbf{x})$, sampled from the base table. Let $\epsilon_t$ be random scalars, independent from $\mathbf{x}$, with $\mathbb{E}[\epsilon_t] = 0$. We define the potential outcomes as

$$Y_t = \mu_t(\mathbf{x}) + \eta_t(\mathbf{x})\,\epsilon_t, \quad t \in \mathcal{T}. \tag{44}$$

This construction preserves the conditional means, that is $\mathbb{E}[Y_t \mid \mathbf{x}] = \mu_t(\mathbf{x})$. The input-dependent scale factors $\eta_t(\mathbf{x})$ allow for heteroscedastic noises and capture a richer family of outcome distributions than additive parametric noise models. For our training, we sample $\epsilon_t$ from a Gaussian with a variance uniformly drawn from $(0, \text{Var}(\mu_t)]$. This choice of noise values ensures a similar noise scale to the scale of CEPOs, resulting in training data with a more informative signal-to-noise ratio.

**Propensities with Positivity Control.** Given a covariate vector $\mathbf{x}$, the strong ignorability assumption requires the propensity values $0 < P(T = 1 \mid \mathbf{X} = \mathbf{x}) < 1$. Hence, due to the invertibility of the sigmoid function, it is sufficient to generate treatment logits, through any function $f : \mathbf{X} \to \mathbb{R}$, and then apply a sigmoid function to get values within $(0, 1)$. To simulate different degrees of confounding, we choose $f$ by randomly selecting one of the following mechanisms:

*(i)* **Randomized treatments (RCT).** Treatments are independent of covariates, i.e., $f$ is constant. We sample $c \sim \text{Logistic}(0, 1)$ and set $f(\mathbf{x}) = c$ to get uniform propensities.

*(ii)* **Linear logits.** Draw the random vector $\mathbf{w}$ from a standard Gaussian and set $f(\mathbf{x}) = \mathbf{w}^\top\mathbf{x}$.

*(iii)* **Non-linear logits.** Feed $\mathbf{x}$ into a randomly initialized MLP, similar architecture to that of Hollmann et al. [42], to get $f(\mathbf{x})$.

Empirically, we observe that the above procedure yields an artificially high level of positivity, which is not reflective of real-world scenarios. We therefore apply a light-weight post-processing transform, inspired by `RealCause` [81], to better control the positivity level. Concretely, we sample a parameter $\xi \in [0, 1]$ and *exacerbate* extreme propensity scores to mimic poor positivity:

$$P(T = 1 \mid \mathbf{X} = \mathbf{x}) := \xi\,\text{Sigmoid}(f(\mathbf{x})) + (1 - \xi)\,\mathbb{I}(f(\mathbf{x}) > 0). \tag{45}$$

Here, $\xi = 1$ leaves the original positivity intact. However, for smaller $\xi$ values, the support of the treated and control groups become increasingly disjoint, leading to low-positivity scenarios.

**Treatment Assignment.** Finally, each unit's treatment is drawn as $T \sim \text{Bernoulli}(\text{Sigmoid}(f(\mathbf{X})))$, and the observed outcome $Y$ is also derived by selecting the assigned potential outcome $Y := Y_T$.

Collectively, all of the steps above simulate different DGPs, with various levels of positivity and heterogeneity, extracted from real and synthetic sources of tabular data. This procedure creates a broad prior $\pi$ for CausalPFN, which is necessary for the model to work well in practice.

## D.2 Model Details

**Architecture & Training.** We represent each context row $(t, \mathbf{x}, y)$ and query row $(t, \mathbf{x})$ as single tokens by summing up (1) a treatment embedding for $t$, (2) a covariate embedding for $\mathbf{x}$ (padded to length $F = 100$), and (3) an outcome embedding for $y$ (only for context rows). We use linear layers for embeddings and omit the positional encodings to preserve the permutation invariance of the context set, similar to other PFN-style transformers.

All tokens—context and query—are passed into a 20-layer transformer, with a hidden size of 384, QK-normalization (RMS)[5], and a parallel SwiGLU-activated [105] feed-forward block.

The transformer's query outputs are then projected to a 1024-dimensional logit vector, then softmaxed at a fixed temperature of $\theta_T = 1.0$ to form a discrete CEPO posterior over the interval $[-10, 10]$. We then scale the interval to match the scale of the outcomes and clip the out-of-range values. At inference time, we return the posterior mean as the point estimate and sample 10,000 times to estimate credible intervals at any desired significance level $\alpha$.

The full model has approximately 20M parameters and is trained in two stages: *(i)* a predictive phase that mimics standard predictive PFN training from Ma et al. [69], and *(ii)* a causal phase that optimizes the CEPO loss. We use AdamW [55] with warmup and cosine annealing for the predictive phase, and switch to the schedule-free optimizer [24] in the causal phase. The model is trained with a maximum context length of 16K in the first phase and 2,048 in the second. We use four A100 GPUs trained for at most one week for the initial phase, and two days on an H100 for the second phase.

Finally, to enhance parallel training, we batch both the queries and the tables. That is, rather than sampling only one DGP and one query token, each gradient update samples $B_t$ DGPs, draws $B_q$ queries per DGP, and concatenates everything into a single tensor. The tensor is then passed through the transformer to get $B_t B_q$ CEPO-PPDs. The final loss is averaged over all the batches. See Algorithm 1 for a detailed demonstration of CausalPFN's training algorithm.

---

**Algorithm 1** Parallel training of CausalPFN.

---

**Require:** Prior $\pi$, DGPs and CEPO values $P_{\text{obs}}^{\psi}$, $\mu_t(\cdot; \psi)$, model $q_\theta$, DGP batch size $B_t$, query batch size $B_q$, fixed feature length $F$, and histogram loss $\texttt{HL}$ (10) [44].

1: **while** not converged **do**
2: $\quad$ Sample $\psi[1], \dots, \psi[B_t] \sim \pi$
3: $\quad$ Sample $\mathcal{D}_{\text{obs}}[i] \sim P_{\text{obs}}^{\psi[i]}, \forall 1 \leq i \leq B_t$
4: $\quad$ Randomly sample query treatments $t^{(i,j)}$ for $1 \leq i \leq B_t, 1 \leq j \leq B_q$
5: $\quad$ Sample query covariates $\mathbf{x}^{(i,j)} \sim P_{\text{obs}}^{\psi}[i]$ for $1 \leq i \leq B_t, 1 \leq j \leq B_q$
6: $\quad$ Set $\mu^{(i,j)} \leftarrow \mu_{t^{(i,j)}} \left( \mathbf{x}^{(i,j)} ; \psi[i] \right)$
7: $\quad$ Pad $\mathbf{x}^{(i,j)}$ with zeros such that $\mathbf{x}^{(i,j)} \in \mathbb{R}^F$
8: $\quad$ $\widehat{\mathcal{L}} \leftarrow \frac{1}{B_t \cdot B_q} \sum_{i,j} \texttt{HL} \left[ \mu^{(i,j)} \| q_\theta(\cdot \mid \mathbf{x}^{(i,j)}, t^{(i,j)}, \mathcal{D}_{\text{obs}}[i]) \right]$
9: $\quad$ Update $\theta$ using the gradients $\nabla_\theta \widehat{\mathcal{L}}$
10: **end while**

---

**Handling Large Tables at Inference Time.** CausalPFN's default maximum context length is set to 4,096 at inference, but real-world tables may contain millions of rows. Training PFN-style transformers on such long contexts can be challenging due to hardware or architectural constraints. While some tabular foundation models such as TabICL [92] modify the architecture itself, Thomas et al. [109] show that, retrieving a small relevant subset of rows for each query at inference time allows a model with a short context length to better generalize to longer contexts.

We adopt this retrieval philosophy in CausalPFN to enable causal effect estimation on large tables. First, we fit a lightweight gradient boosting regressor on the context data to produce weak CATE estimates for each covariate. This regressor estimates CATE by regressing outcomes on the treatment and covariates and then taking the difference in predicted outcomes between $T = 1$ and $T = 0$. This step is applied *only* when the table is too large to fit within the model's maximum context window. We then (i) sort both the context rows and the queries based on their weak CATE estimates, which

---

[5]Different from Henry et al. [38], we perform normalization *after* the query and key projection.

effectively stratifies the data; (ii) partition the ordered queries into consecutive mini-batches; and (iii) for each query batch, use a fast bisection search to select a contiguous window of context rows whose weak CATE estimate range most closely matches that of the batch. As a result, each batch is exposed only to a neighborhood of rows with similar causal effects, allowing all CEPO predictions to be computed with short forward passes.

## D.3 Sensitivity to Dataset Size

During the causal phase of training, we consider sample sizes up to 2,048 and covariates up to 100. However, during inference, CausalPFN can take up to 50,000 samples.

To assess the effect of context size and dimensionality on CausalPFN's performance, we run additional experiments on synthetic polynomial datasets. The test set size is fixed at 100 in all experiments. For each *(rows, covariates)* configuration, we report mean $\pm$ standard error over 50 datasets drawn from the polynomial prior with different random seeds.

**Effect of Sample Size.** We consider the same DGPs while increasing the number of samples and fixing the number of covariates to 10. Table 4 reports PEHE for CATE across baselines. CausalPFN exhibits faster PEHE decay with increasing rows, with a slight plateau at very large contexts.

Table 4: Effect of sample size on PEHE (mean $\pm$ SE). Covariates = 10; averages over 50 datasets.

| Method | Number of Rows | | | | | | | |
|---|---|---|---|---|---|---|---|---|
| | 10 | 20 | 50 | 100 | 200 | 500 | 5,000 | 10,000 |
| CausalPFN | 1.34±0.02 | 1.27±0.02 | 1.10±0.02 | 0.89±0.02 | 0.74±0.03 | 0.46±0.01 | 0.29±0.01 | 0.31±0.01 |
| DA-Learner | 1.33±0.02 | 1.30±0.02 | 1.16±0.01 | 1.00±0.01 | 0.91±0.03 | 0.85±0.01 | 0.84±0.02 | 0.82±0.02 |
| S-Learner | 1.44±0.01 | 1.40±0.02 | 1.35±0.02 | 1.21±0.02 | 1.18±0.05 | 1.07±0.03 | 1.00±0.04 | 1.03±0.04 |
| T-Learner | 1.33±0.02 | 1.30±0.02 | 1.15±0.01 | 0.97±0.01 | 0.88±0.02 | 0.81±0.01 | 0.81±0.02 | 0.81±0.02 |
| X-Learner | 1.35±0.02 | 1.32±0.02 | 1.20±0.02 | 1.04±0.01 | 0.94±0.03 | 0.87±0.01 | 0.87±0.02 | 0.84±0.02 |

**Effect of Covariate Size.** Next, we fix the number of samples to 1,000 and vary the number of covariates. Table 5 compares CausalPFN's performance to other methods in terms of PEHE. Although CausalPFN consistently outperforms the baselines, the performance gap narrows as the number of covariates grows—likely due to training exposure being limited to up to 100 dimensions, which could be mitigated by training with higher-dimensional inputs.

Table 5: Effect of covariate size on PEHE (mean $\pm$ SE). Samples = 1,000; averages over 50 datasets.

| Method | Number of Covariates | | | | | | | |
|---|---|---|---|---|---|---|---|---|
| | 1 | 5 | 10 | 20 | 50 | 100 | 500 | 1,000 |
| CausalPFN | 0.08±0.00 | 0.17±0.01 | 0.40±0.01 | 0.67±0.01 | 0.87±0.02 | 1.01±0.02 | 1.28±0.02 | 1.32±0.02 |
| DA-Learner | 0.21±0.01 | 0.59±0.01 | 0.85±0.01 | 1.04±0.01 | 1.14±0.01 | 1.22±0.01 | 1.30±0.02 | 1.32±0.02 |
| S-Learner | 0.54±0.04 | 0.83±0.02 | 1.09±0.02 | 1.17±0.02 | 1.21±0.02 | 1.23±0.01 | 1.30±0.02 | 1.33±0.02 |
| T-Learner | 0.27±0.01 | 0.60±0.01 | 0.83±0.01 | 0.98±0.01 | 1.09±0.01 | 1.18±0.01 | 1.28±0.02 | 1.32±0.02 |
| X-Learner | 0.29±0.01 | 0.64±0.01 | 0.88±0.01 | 1.06±0.01 | 1.15±0.01 | 1.23±0.01 | 1.30±0.02 | 1.33±0.02 |

## D.4 Discussion on Inference Speed

Many applied settings prioritize throughput and latency over marginal gains in asymptotic accuracy. Real-time bidding must estimate incremental ad effects and decide bids within strict millisecond budgets [119]. Likewise, e-commerce personalization depends on rapid uplift estimation within short user sessions, where serving latency directly affects conversion [101].

Although CausalPFN requires substantial offline training, it is designed for zero-shot deployment on new tables with no test-time fitting or adaptation. At inference, interventional queries reduce to a small and fixed number of forward passes, and the computation parallelizes well across large batches (e.g., using mixed precision, caching).

Accordingly, Figure 1 does not claim that CausalPFN is intrinsically faster than every baseline; rather, it reflects practitioner-facing wall-clock time from data arrival to effects returned. Baselines that require per-dataset refitting or tuning incur this cost at deployment, whereas CausalPFN does not.

## D.5 Baseline Hyperparameters and Results without Hyperparameter Tuning

**No Hyperparameter Tuning.** Table 6 summarizes the performance of all methods without hyperparameter tuning. CausalPFN attains the best (second-best) average rank on CATE (ATE).

**EconML Hyperparameters.** For the results without hyperparameter tuning in Table 6, we ran the models with the recommended hyperparameters in the Jupyter notebooks from EconML [11]. For the tuned results in Table 1, we performed hyperparameter tuning using the FLAML (AutoML) library [113] on both the propensity and outcome models with *(i)* Time budget of 900 seconds, *(ii)* K-fold cross-validation with $K = 3$, *(iii)* Early stopping, and *(iv)* base estimators ["lgbm", "xgboost", "xgb_limitdepth", "rf", "kneighbor", "extra_tree", "lrl1", "lrl2"]. For Forest DR-Learner and Forest DML, we additionally expanded the covariates with cubic terms (polynomial degree 3), with an additional tuning of the final model.

**CATE Nets.** For the results without hyperparameter tuning in Table 6, we ran the models with the default hyperparameters and a batch size of 512. For the tuned results in Table 1, we perform a grid search on the hyperparameters for the neural architecture: *(i)* Number of layers $\in \{2, 3\}$, *(ii)* Representation dimension $\in \{128, 256\}$, *(iii)* Number of hidden output layers $\in \{1, 2\}$, and *(iv)* Width of the hidden output layers $\in \{128, 256\}$. The rest of the hyperaparameters are left unchanged.

**BART & GRF.** The GRF implementation includes an internal `tune` option. We enable this option in Table 1 and disable it for the untuned experiment in Table 6. BART, on the other hand, offers no comparable hyperparameter-tuning. Its only alternative, a full cross-fit, is prohibitively slow and uses a rudimentary Bayesian routine. Thus, the BART scores appear unchanged in Tables 1 and 6.

Table 6: **CATE & ATE results.** PEHE *(left half)* alongside ATE relative error and its overall average *(right half)*. PEHE for Lalonde CPS/PSID is shown in thousands. Best numbers are in **blue**; second best are in **purple**. Cells with "—" indicate that the method is not applicable.

| Method | Mean PEHE ± Standard Error (↓ better) | | | | | Mean ATE Relative Error ± Standard Error (↓ better) | | | | |
|---|---|---|---|---|---|---|---|---|---|---|
| | IHDP | ACIC 2016 | Lalonde CPS ($\times 10^3$) | Lalonde PSID ($\times 10^3$) | Avg. Rank | IHDP | ACIC 2016 | Lalonde CPS | Lalonde PSID | Avg. Rank |
| **CausalPFN** | 0.58±0.07 | 0.92±0.11 | 8.96±0.02 | 14.40±0.20 | 2.17±0.09 | 0.20±0.04 | 0.05±0.01 | 0.13±0.01 | 0.22±0.02 | 4.26±0.18 |
| DA-Learner | 2.98±0.51 | 1.88±0.24 | 9.01±0.02 | 13.96±0.19 | 3.64±0.18 | 0.22±0.04 | 0.09±0.03 | 0.22±0.01 | 0.08±0.01 | 4.15±0.19 |
| T-Learner | 2.94±0.49 | 2.06±0.20 | 9.29±0.02 | 13.91±0.18 | 4.01±0.18 | 0.22±0.04 | 0.11±0.03 | 0.40±0.01 | 0.07±0.01 | 4.62±0.18 |
| DragonNet | 2.13±0.24 | 2.23±0.20 | 10.83±0.15 | 16.40±0.27 | 5.62±0.17 | 0.21±0.04 | 0.09±0.02 | 0.56±0.03 | 0.44±0.02 | 6.04±0.17 |
| IPW | — | — | — | — | — | 0.23±0.04 | 0.24±0.05 | 0.22±0.01 | 0.07±0.01 | 4.33±0.20 |
| TarNet | 1.89±0.15 | 2.26±0.20 | 12.00±0.04 | 18.71±0.16 | 6.87±0.11 | 0.21±0.04 | 0.06±0.02 | 0.90±0.01 | 0.72±0.01 | 7.54±0.14 |
| X-Learner | 3.70±0.62 | 1.71±0.31 | 12.28±0.03 | 21.72±0.16 | 8.13±0.16 | 0.19±0.03 | 0.07±0.02 | 0.83±0.01 | 0.92±0.01 | 7.92±0.17 |
| RA-Net | 2.08±0.19 | 2.42±0.22 | 12.86±0.12 | 20.13±0.41 | 8.18±0.16 | 0.20±0.04 | 0.07±0.03 | 0.96±0.02 | 0.71±0.04 | 7.95±0.17 |
| BART | 2.50±0.39 | 0.68±0.11 | 12.81±0.05 | 21.36±0.16 | 8.20±0.17 | 0.44±0.09 | 0.04±0.01 | 0.99±0.01 | 0.86±0.01 | 8.72±0.18 |
| GRF | 4.26±0.69 | 1.36±0.30 | 12.18±0.06 | 21.84±0.16 | 8.21±0.17 | 0.18±0.03 | 0.07±0.02 | 0.81±0.02 | 0.85±0.02 | 7.78±0.17 |
| S-Learner | 3.91±0.68 | 2.23±0.28 | 12.88±0.02 | 22.68±0.13 | 9.29±0.18 | 0.28±0.05 | 0.12±0.05 | 1.00±0.00 | 1.03±0.00 | 9.99±0.18 |
| Forest DR Learner | 3.90±0.66 | 1.68±0.35 | 26.08±4.96 | 22.55±0.25 | 9.51±0.18 | 0.19±0.04 | 0.08±0.04 | 1.39±0.28 | 0.87±0.03 | 8.35±0.17 |
| Forest DML | 4.40±0.72 | 1.47±0.32 | 15.12±0.15 | 23.12±0.15 | 10.51±0.18 | 0.09±0.02 | 0.05±0.02 | 1.12±0.02 | 1.02±0.01 | 9.37±0.23 |

All inference, including baselines, performed on an 80 GB H100 GPU, 32 CPUs, and 256 GB RAM.

## D.6 Marketing Experiments

**Datasets.** Apart from Hill$^{(1)}$ and Hill$^{(2)}$, which were explained in the main text. We also run experiments on the following datasets:

1. **Criteo.** 25M ad-exposure records from Criteo's online *incrementality tests*: a randomly selected *held-out* audience is shielded from seeing an advert, while the treated audience is shown the ad; the target is a post-impression conversion flag. We use a readily provided 2.5M stratified subset of this dataset from `sklift`.

2. **Retail-Hero (X5).** Transaction logs from the X5 Retail Group hackathon. Customers are randomly offered personalized coupons (treatment); the outcome records whether the customer subsequently purchased the promoted items.

3. **Lenta.** SMS-based promotion experiment run by the grocery chain Lenta. The treatment group receives a marketing text, and the outcome is a visit after the campaign window.

4. **Megafon (Mega).** Synthetic yet domain-faithful data released for the MegaFon Uplift competition. Users are randomly offered a telecom upsell offer (treatment), and the outcome indicates whether they accepted the offer.

**Qini Evaluation.** To build Qini curves we follow `scikit-uplift`'s recommended five-fold *stratified* split based on the outcome and the treatment [74]. In each fold, we hold out 20% of the data as test rows and train the baseline models on the remaining 80%. For CausalPFN we use that same 80% as context tokens and treat the held-out 20% as queries. We then rank the rows based on their CATE estimates to compute the Qini curves and the corresponding Qini scores.

**Context Length Challenges.** In all the marketing experiments, we have increased the model's maximum context length from the default 4,096 to 50,000 tokens. This context length is sufficient for the subsampled datasets in Table 2. However, extending beyond 50K for the *full-table* runs is not feasible in GPU memory. We thus use the retrieval approach explained in Appendix D.2 to achieve CATE estimates for this setting. Table 7 shows CausalPFN's performance (with the retrieval approach) compared to the baselines on the full-table datasets. We conjecture that the relative under-performance compared to Table 2 is due to this retrieval heuristic.

Table 7: **Normalized Qini scores** (↑ better). Scores are normalized per dataset such that the top-performing model achieves 1.0 (highlighted in **bold**). All datasets use full stratified subsamples: $\text{Hill}^{(1)}$ and $\text{Hill}^{(2)}$ (64K rows), Criteo (2.5M rows), X5 (200K rows), Lenta (687K rows), and Mega (600K rows).

| Method | $\text{Hill}^{(1)}$ | $\text{Hill}^{(2)}$ | Criteo | X5 | Lenta | Mega | Avg. |
|---|---|---|---|---|---|---|---|
| S Learner | **1.000** | **1.000** | **1.000** | **1.000** | **1.000** | 0.913 | **0.985** |
| X Learner | 0.975 | 0.980 | 0.994 | 0.965 | 0.868 | 0.997 | 0.963 |
| DA Learner | 0.985 | 0.964 | 0.955 | 0.969 | 0.903 | **1.000** | 0.963 |
| T Learner | 0.991 | 0.972 | 0.902 | 0.953 | 0.833 | 0.987 | 0.940 |
| CausalPFN | 0.992 | 0.968 | 0.939 | 0.746 | 0.947 | 0.954 | 0.924 |

## D.7 Calibration, Coverage, and Credible Intervals

**The Synthetic DGPs.** For the calibration results in Figure 7, we use two families of synthetic DGPs, polynomials and sinusoidals. As a general recipe, each DGP defines a treatment logit function $f(\mathbf{x}) \in \mathbb{R}$ and assigns treatments by sampling from the $\text{Bernoulli}(\text{Sigmoid}(f(\mathbf{x})))$. Moreover, each DGP specifies two CEPO functions $\mu_0, \mu_1 : \mathcal{X} \to \mathbb{R}$. It then samples the potential outcomes by $y_t = \mu_t(\mathbf{x}) + \epsilon_t$ for $t \in \{0, 1\}$, where the noise terms $\epsilon_t \sim \text{Normal}(0, 1)$, $\text{Laplace}(0, 1)$, or $\text{Uniform}(-1, 1)$ with equal probability. We now describe each DGP family in more detail:

(a) **Polynomial.** We first draw the number of features $d \sim \text{Unif}\{10, \ldots, 20\}$ and sample covariate vectors $\mathbf{x} \sim \text{Unif}[-2, 2]^d$. We then fix a maximum degree $\deg \in \{1, 2, 3, 4\}$, augment covariates with powers $\mathbf{x}_{\text{ext}} = (x_1, \ldots, x_d, x_1^2, \ldots, x_d^{\deg})$, sample weights $\mathbf{w}_{\mu_0}, \mathbf{w}_{\mu_1}, \mathbf{w}_T \sim \text{Unif}[-5, 5]^{d \times \deg + 1}$, and define

$$f(\mathbf{x}) = \mathbf{w}_T^\top \mathbf{x}_{\text{ext}}, \quad \mu_t(\mathbf{x}) = \mathbf{w}_{\mu_t}^\top \mathbf{x}_{\text{ext}} \text{ for } t \in \{0, 1\}. \tag{46}$$

Degrees 1, 2, 3, and 4 give the Linear, Quadratic, Cubic, and Quartic sub-families; each degree adds new terms and is therefore a super-set of all lower degrees. We train on one degree family and test on the others to probe generalization.

(b) **Sinusoidal.** We draw the number of features $d \sim \text{Unif}\{5, \ldots, 10\}$ and sample covariate vectors $\mathbf{x} \sim \text{Unif}[-3, 5]^d$. We then sample weight vectors $\mathbf{w}_{\mu_0}, \mathbf{w}_{\mu_1}, \mathbf{w}_T \sim \text{Unif}[-10, 6]^d$, and a frequency $\omega \in \mathbb{R}^+$. We define the treatment logit function and the CEPOs as

$$f(\mathbf{x}) = \sin\left(\omega \left\{\mathbf{w}_T^\top \mathbf{x}\right\}\right) + \mathbf{w}_T^\top \mathbf{x}, \quad \mu_t(\mathbf{x}) = \sin\left(\omega \left\{\mathbf{w}_{\mu_t}^\top \mathbf{x}\right\}\right) + \mathbf{w}_{\mu_t}^\top \mathbf{x} \text{ for } t \in \{0, 1\}. \tag{47}$$

For training DGPs, we create three sub-families: Linear ($\omega = 0$), L1 ($\omega \in [0, 1]$) and L2 ($\omega \in (1, 2]$). For test-time DPGs, we use the following: Linear ($\omega = 0$), L1 ($\omega \in [0.5, 1]$), L2 ($\omega \in (1.5, 2]$), and L3 ($\omega \in (2.5, 3]$). This allows us to measure extrapolation to unseen frequencies. For example, an L2-trained model has seen DGPs from L1 and L2, but not L3.

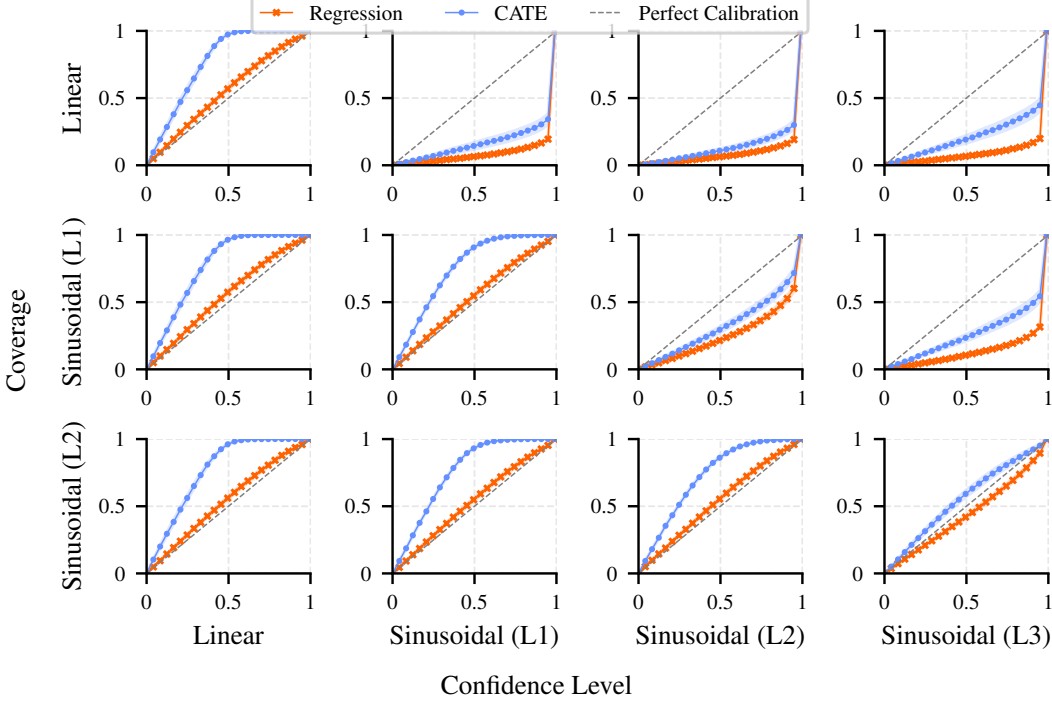

Figure 8: CATE and regression calibration curves for synthetic sinusoidal datasets, **before calibration.** Models are trained on Linear/Sinusoidal (L1)/Sinusoidal (L2) datasets and tested on Linear/Sinusoidal (L1)/Sinusoidal (L2)/Sinusoidal (L3) benchmarks.

**Synthetic Experiments on Sinusoidal.** Figure 8 shows both the regression curve $\widehat{cov}_\mu$ (orange) and the CATE curve $\widehat{cov}_\tau$. The model is overly confident in OOD scenarios (e.g., L2 tested on an L1 trained model) and either well-calibrated or conservative otherwise. The figure also shows that the regression curve is always below the blue CATE curve. Once calibration is done on the regression curve, as shown in Figure 9, the $\text{ICE}_\mu$ becomes smaller, resulting in a well-calibrated or conservative model, even on OOD scenarios.

**Synthetic Experiments on Polynomial.** Similar to the sinusoidal setting, the uncalibrated curves in Figure 10 show that the model becomes overly confident when tested on OOD data (e.g., testing a model trained on Quadratic data on Cubic DGP). However, applying the regression calibration results in near-perfect CATE calibration, as shown in Figure 11.

**Calibration of the Large-scale CausalPFN.** We evaluate the calibration curves of the large-scale pre-trained CausalPFN on both synthetic and standard benchmarks in Figures 12 to 14. The model generally appears conservative. This may be attributed to the Gaussian smoothing used in the histogram loss; yet, this smoothing is necessary to achieve stability in training. Regardless, across all datasets, post-hoc regression calibration improves reliability: the calibrated (pink) curves adhere far more closely to the diagonal than their uncalibrated (blue) counterparts. In Figures 12 and 13 the improvement is almost perfect, while in Figure 14 it corrects the base model's strong conservatism on IHDP and ACIC 2016 and achieves near-ideal alignment on the Lalonde datasets.

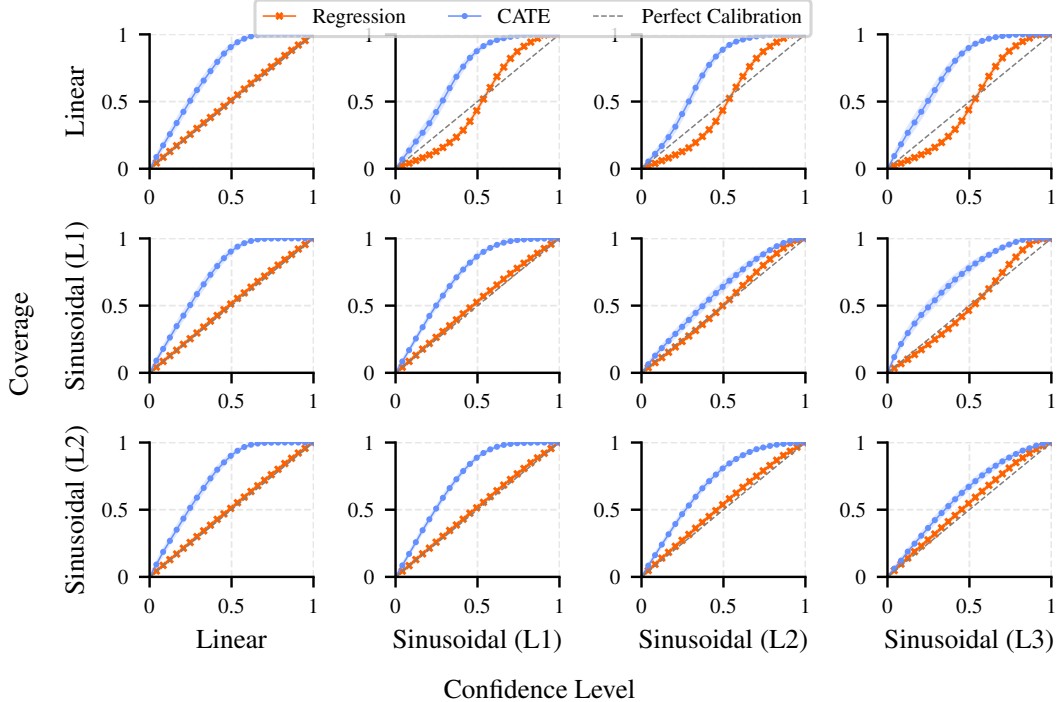

Figure 9: CATE and regression calibration curves for synthetic sinusoidal datasets, **after calibration.** Models are trained on Linear/Sinusoidal (L1)/Sinusoidal (L2) datasets and tested on Linear/Sinusoidal (L1)/Sinusoidal (L2)/Sinusoidal (L3) benchmarks.

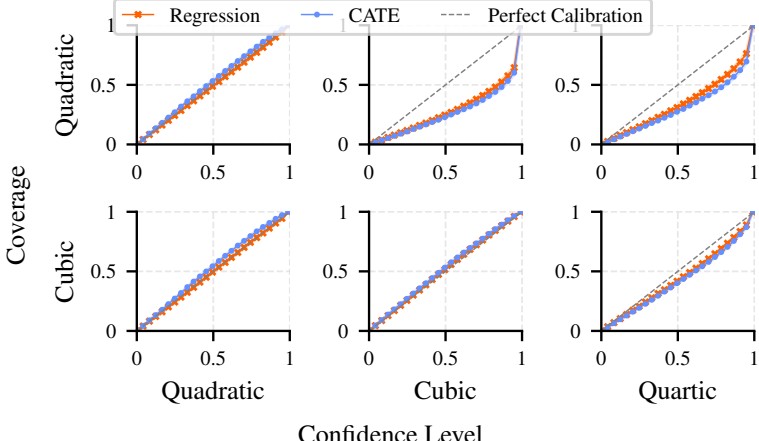

Figure 10: CATE and regression calibration curves for synthetic polynomial datasets, **before calibration.** Models are trained on Quadratic/Cubic datasets and tested on Quadratic/Cubic/Quartic ones.

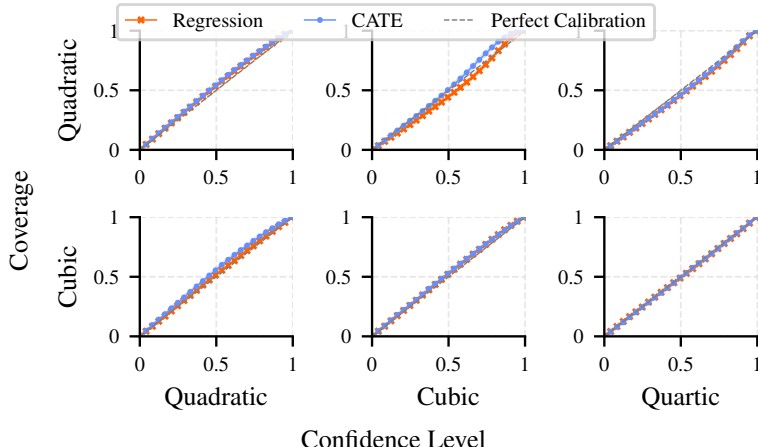

Figure 11: CATE and regression calibration curves for synthetic polynomial datasets, **after calibration.** Models are trained on Quadratic/Cubic datasets and tested on Quadratic/Cubic/Quartic ones.

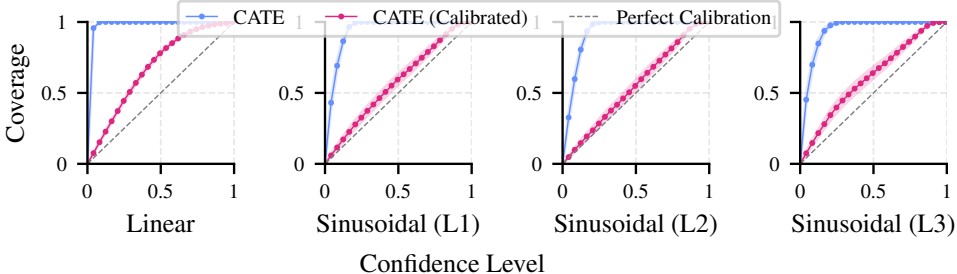

Figure 12: CausalPFN's CATE calibration on sinusoidal datasets, **before and after calibration.**

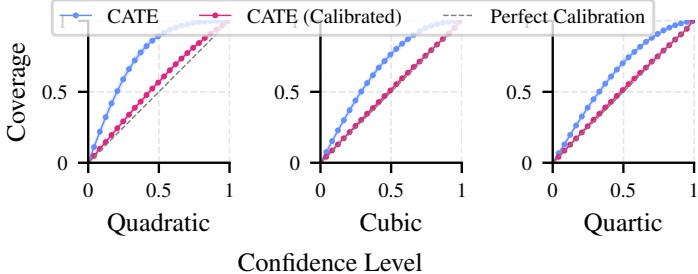

Figure 13: CausalPFN's CATE calibration on polynomial datasets, **before and after calibration.**

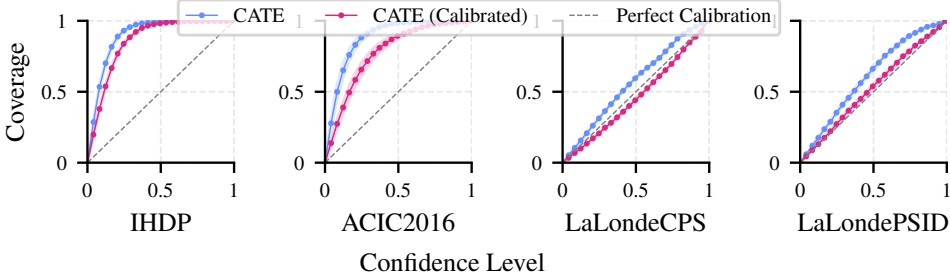

Figure 14: CausalPFN's CATE calibration on standard benchmarks, **before and after calibration.**

# E  Concurrent Work on PFNs for Causal Inference

Do-PFN [96] is a concurrent approach that extends TabPFN to interventional queries by learning the interventional posterior predictive distribution, i.e., a distribution over $Y_t$ given $(\mathbf{X}, \mathcal{D}_{\text{obs}})$. In contrast, CausalPFN targets the *expectation* of the interventional distribution (e.g., $\mathbb{E}[Y_t \mid \mathbf{X}{=}\mathbf{x}]$), thus removing outcome (aleatoric) noise from the prediction target. This is especially relevant for uncertainty quantification, where we aim to isolate epistemic uncertainty about the causal effect.

More importantly, as described, Do-PFN does not explicitly enforce identifiability: the training prior can include *observationally equivalent* DGPs (distinct processes with the same $P(\mathbf{X}, T, Y)$ but different effects). As formalized in Proposition 1, if the training prior admits such cases, then any learner that conditions only on observational data cannot, in general, have its posterior predictive concentrate on the true effect, even with unlimited samples and model capacity. CausalPFN avoids this by constructing a prior that satisfies the ignorability (identifiability) condition, ensuring that CEPOs are functionals of $P_{\text{obs}}$ (one effect per observational law). Empirically, CausalPFN outperforms Do-PFN on standard benchmarks in both PEHE and ATE relative error (Table 8).

Table 8: Head-to-head comparison on benchmarks (mean $\pm$ SE; $\downarrow$ is better). For PEHE, Lalonde CPS/PSID values are reported $\times 10^3$.

|  |  | IHDP | ACIC 2016 | Lalonde CPS | Lalonde PSID |
|---|---|---|---|---|---|
| **PEHE** ($\downarrow$) | CausalPFN | $0.58 \pm 0.07$ | $0.92 \pm 0.11$ | $8.96 \pm 0.02$ | $14.40 \pm 0.20$ |
|  | Do-PFN | $6.07 \pm 0.89$ | $4.11 \pm 0.52$ | $12.01 \pm 0.03$ | $20.91 \pm 0.14$ |
| **ATE Relative Error** ($\downarrow$) | CausalPFN | $0.20 \pm 0.04$ | $0.05 \pm 0.01$ | $0.13 \pm 0.01$ | $0.22 \pm 0.02$ |
|  | Do-PFN | $0.57 \pm 0.10$ | $0.67 \pm 0.04$ | $0.87 \pm 0.01$ | $0.92 \pm 0.01$ |

