# OpenReview forum: "CausalPFN: Amortized Causal Effect Estimation via In-Context Learning"
_NeurIPS.cc/2025/Conference — NeurIPS 2025 spotlight_

### Official Review · Reviewer_URzh · 2025-06-25

**Clarity:** 3
**Significance:** 3
**Originality:** 3
**Rating:** 5
**Confidence:** 4

**Summary:**

PFN is a method that trains a Transformer to input a training dataset (X_train, y_train) and input data X_test for inference, and then outputs the inference result y_test. The authors applied this method to causal effect inference and proposed a method for creating a set of training datasets that satisfy strong ignorability assumption. Specifically, two columns of the explanatory variables in the data are designated as latent outcomes (Y0, Y1), and a random propensity score is used to generate T from the remaining variables, with the corresponding latent outcomes Y_T designated as factual outcome Y.
The learned model achieves higher accuracy than several existing methods and is also faster (benefiting from the PFN-type model where the learner is pre-trained).

**Questions:**

[Q1] Since quite high accuracy is reported for IHDP CATE and Lalonde CPS ATE, is there any concern that the pre-training data might contain these datasets? How is it possible to confirm that such dataset leaks have not occurred?

[Q2] Is it possible to train a model that directly estimates CATE instead of CEPO? That is, a transformer that takes (X_train, T_train, Y_train, X_test) as input and outputs τ_test? In causal inference, it is important to focus on the causal effect τ itself, which is often relatively smaller than the variation of Y on X (since we focus only on interventionable factors as $t$). The proposed method selects Y1 and Y0 randomly from X, which does not reproduce such a situation.

**Ethical Concerns:**

["NO or VERY MINOR ethics concerns only"]

**Final Justification:**

The main concern regarding the outdated baselines is properly addressed, adding FlexTENet and SNet.

**Limitations:**

A limited amount of training data is inherent in the PFN approach, which is properly discussed.

**Paper Formatting Concerns:**

No major concern is found.

**Quality:**

2

**Strengths And Weaknesses:**

## Strengths

It is significant to demonstrate that the PFN approach is also effective for causal inference.

## Weaknesses

[W1] Baselines are somewhat outdated, and thus, the benefit in performance is unclear. Although they used the CATENets library, they did not compare SOTA DNN-based methods in it, such as SNet and FlexTENet.

[W2] Causal inference usually involves statistical performance issues, and computational performance is not particularly important in practical terms. The advantage at this point may not be appealing to readers.
Therefore, speed alone is not a compelling reason for adoption.
Authors might want to add some explanations of use-cases where speed is important, or of the value added by high-speed inference.

---

> ### Author Rebuttal · Authors · 2025-07-31
>
> We thank the reviewer for their thoughtful review and constructive feedback. Below, we have provided detailed responses to each of the concerns raised. If our clarifications address these points satisfactorily, we would be very grateful if you consider raising the score.
>
>
> > Baselines are somewhat outdated, and thus, the benefit in performance is unclear. Although they used the CATENets library, they did not compare SOTA DNN-based methods in it, such as SNet and FlexTENet.
>
> We have used two widely-used libraries for heterogeneous treatment effect estimation: EconML (4.2k stars on Github), and CATENets (another widely used library), in addition to more classical R-based baselines like GRF and BART. Moreover, we tuned the hyperparameters carefully for most of the baselines. Based on the reviewer’s suggestion, we also provide additional baselines from the CATENets library, namely SNet, FlexTENet, as well as XNet.
>
> Here, we compare CausalPFN to these additional baselines on the IHDP, ACIC, and Lalonde datasets for both ATE estimation (relative error) and CATE estimation (PEHE). We perform hyperparameter tuning similar to that used for the other baselines in the CATENets library, and still show competitive performance often exceeding that of these new baselines.
>
> **1) Mean PEHE ± Standard Error (↓ better)**
>
> | Method | IHDP | ACIC 2016 | Lalonde CPS (×10³) | Lalonde PSID (×10³) |
> |---|--|---|----|---|
> | **CausalPFN** | **0.58±0.07** | **0.92±0.11** | **8.96±0.02** | **14.40±0.20** |
> |SNet|2.12±0.13 | 1.46±0.17 |12.27±0.09 | 15.42±0.45|
> |FlexTENet|2.08±0.11|2.30±0.26|10.33±0.10| 17.18±0.73|
> |XNet|3.02±0.34| 2.34±0.26| 12.11±0.18    | 20.22±1.00  |
>
> **2) Mean ATE Relative Error ± Standard Error (↓ better)**
>
> | Method | IHDP | ACIC 2016 | Lalonde CPS | Lalonde PSID |
> |---|---|----|---|--|
> | **CausalPFN** | 0.20±0.04| 0.05±0.01 | **0.13±0.01** | **0.22±0.02** |
> |FlexTENet|0.21±0.04    |**0.04±0.01**    |0.53±0.02    |0.40±0.06|
> |SNet|0.21±0.04    |**0.04±0.01**    |0.79±0.02| 0.25±0.03|
> |XNet|**0.20±0.03**|0.06±0.02    |0.63±0.03|0.49±0.08|
>
>
> > Causal inference usually involves statistical performance issues, and computational performance is not particularly important in practical terms. The advantage at this point may not be appealing to readers. Therefore, speed alone is not a compelling reason for adoption. Authors might want to add some explanations of use-cases where speed is important, or of the value added by high-speed inference.
>
> First, we note that speed is not the main advantage of CausalPFN. As shown in Figure 1, as well as Tables 1 and 2, CausalPFN provides superior performance (for CATE estimation) on average, even without taking the computational performance into account.
>
> Second, while it is true that many traditional causal inference applications prioritize statistical performance, a large number of real-world industry use-cases require fast/adaptive causal estimators. For instance, real-time bidding systems must estimate the incremental effect of ads and make bidding decisions with minimum latency [A]. Another example is e-commerce personalization systems that rely on fast uplift models to tailor recommendations in the course of a short user session, where latency directly impacts conversion rates [B].
>
> Therefore, even though speed is not a primary advantage of our method, we find it to be beneficial in many practical scenarios. We will add further clarifications and add some discussion to further clarify this in our revision.
>
> [A] Yuan et al. "Real-time bidding for online advertising: measurement and analysis." Proceedings of the seventh international workshop on data mining for online advertising. 2013.
>
> [B] Sawant et al. "Contextual multi-armed bandits for causal marketing." arXiv:1810.01859.
>
>
> > [Q1] Since quite high accuracy is reported for IHDP CATE and Lalonde CPS ATE, is there any concern that the pre-training data might contain these datasets? How is it possible to confirm that such dataset leaks have not occurred?
>
> This is quite an important concern for all large-scale training paradigms, including CausalPFN. We made sure that none of the benchmarking datasets was part of the pre-training data based on the following:
>
> 1. Most of the pre-training data used in our model are purely synthetic, following a similar approach of using randomly generated MLPs as in TabPFN v1 [42]. Specifically, no information regarding the benchmarks is used during this synthetic generation.
>
> 2. For the real-world part of the pre-training data, we only use real-world covariates and generate the treatments through synthetic functions and add synthetic noise to the CEPO columns to generate potential outcomes. In that sense, there is only a chance for covariate leakage. However, we have also published the list of real-world datasets (OpenML datasets) in footnote 2 on page 28 of the appendix, which shows that none of the benchmarks are part of those datasets.
>
> > [Q2] Is it possible to train a model that directly estimates CATE instead of CEPO? That is, a transformer that takes (X_train, T_train, Y_train, X_test) as input and outputs τ_test? In causal inference, it is important to focus on the causal effect τ itself, which is often relatively smaller than the variation of Y on X (since we focus only on interventionable factors as ). The proposed method selects Y1 and Y0 randomly from X, which does not reproduce such a situation.
>
>
> Yes, our framework allows us to train CausalPFN to directly estimate CATE values instead of separate estimation of CEPO values $\mathbb{E}[Y_1 | X]$ and $\mathbb{E}[Y_0|X]$. We chose to estimate CEPOs instead of CATE values mainly for the following reason: when calibrating the estimator, we often do not have access to ground-truth CATE values but factual observations, which can correspond to $Y_0$ or $Y_1$ based on the treatment value. Therefore, with the current choice, we can use the observational data to calibrate the model, as we detail in lines 271-307. Ultimately, estimating CEPOs can easily lead to many well-known causal estimands, such as CATE, ATE, local average treatment effects, and can potentially generalize to multiple-treatment settings.

---

> > ### Comment · Reviewer_URzh · 2025-08-02
> > **Concerns properly addressed**
> >
> > I appreciate the authors' dedicated efforts in addressing my concerns. The main concern regarding the outdated baselines is properly addressed, adding FlexTENet and SNet. Other minor issues were also (said to be) handled. I raised my score.

---

### Official Review · Reviewer_8PoL · 2025-06-26

**Clarity:** 3
**Significance:** 3
**Originality:** 3
**Rating:** 5
**Confidence:** 4

**Summary:**

The paper proposes and clearly evaluates a scalable amortized approach for causal effect estimation. The high level idea of using amortized inference has been very successfully applied in causal discovery and this work is one of a couple of concurrent works which explore this idea for cause effect estimation.

The paper is very nicely written, experiments are extensive. The combination of amortization and PFNs makes sense and should be explored. This will be a tool which hopefully can make it into the toolbox of practitioners.

**Questions:**

Why do you only gather identifiable DAGs? Would you not want the distribution over all equivalent DAGs no matter if they are identifiable or not? This would probably give you less clean estimates but would that not better? I.e. right now the argument seems that you only care about a DAG if it is identifiable while in the real world that might not be the case.

Why do you only provide inference code and not training code?

**Ethical Concerns:**

["NO or VERY MINOR ethics concerns only"]

**Final Justification:**

I have nothing to add to my review. It is a good paper and should be accepted.

**Limitations:**

The theoretical framework for causal inference from observational data is unfortunately relatively weak but I do not see much that the authors can do about it.

Overall if training code and the PyPy package is fully released I think it is a pretty good framework to build on top e.g. including interventions, other treatment forms etc.

**Paper Formatting Concerns:**

The paper is nicely formatted.

**Quality:**

3

**Strengths And Weaknesses:**

[1,2,3] are very closely related papers. Some even doing causal inference via PFNs similar to the current work. While all of these papers have been very recent (most of them actually only available after the NeurIPS deadline), it might still be good to discuss these approaches at least or potentially even compare e.g. to [3]. I am already giving the current paper a very high score so I am aware that prior to the submission the authors could not even have known the papers.


[1]Bynum, L. E., Puli, A. M., Herrero-Quevedo, D., Nguyen, N., Fernandez-Granda, C., Cho, K., & Ranganath, R. (2025). Black box causal inference: Effect estimation via meta prediction. arXiv preprint arXiv:2503.05985.

[2]Robertson, J., Reuter, A., Guo, S., Hollmann, N., Hutter, F., & Schölkopf, B. (2025). Do-PFN: In-context learning for causal effect estimation. arXiv preprint arXiv:2506.06039.

[3] Ma, Y., Frauen, D., Javurek, E., & Feuerriegel, S. (2025). Foundation Models for Causal Inference via Prior-Data Fitted Networks. arXiv preprint arXiv:2506.10914.

---

> ### Author Rebuttal · Authors · 2025-07-31
>
> We thank the reviewer for their positive feedback and encouraging comments. We are glad they found the paper well-written, the experiments extensive, and the combination of amortization and PFNs promising. We also appreciate their view that this work could become a valuable tool for practitioners.
>
> > [1,2,3] are very closely related papers. Some even doing causal inference via PFNs similar to the current work. While all of these papers have been very recent (most of them actually only available after the NeurIPS deadline), it might still be good to discuss these approaches at least or potentially even compare e.g. to [3].
>
> Thank you for pointing out these related works. Except for [1], which is already discussed in the paper as reference [14], the other papers first became public on arxiv approximately **one month after the conference submission deadline**. Among the mentioned papers, [3] does not provide any code for using their model and mainly benchmarks on synthetic datasets that we do not have access to. Similarly, we could not find any code for the method in [1]. Specifically, as mentioned in Section 3.6 (page 9) of [1], the method requires having access to the underlying family of data-generating processes for each benchmark (e.g., for IHDP, they use 13,000 training datasets sampled from the underlying outcome surface model). In contrast, CausalPFN provides a zero-shot estimator. Below, we compare to Do-PFN [2], which is the most similar method to ours.
>
> We use the IHDP, ACIC, and Lalonde datasets for both ATE estimation (relative error) and CATE estimation (PEHE). Note that Do-PFN is also a zero-shot method so we do not run any hyperparameter tuning or pre-processing on the datasets.
>
> **1) Mean PEHE ± Standard Error (↓ better)**
>
> | Method | IHDP | ACIC 2016 | Lalonde CPS (×10³) | Lalonde PSID (×10³) |
> |----|---|---|---|--|
> | **CausalPFN** | **0.58±0.07** | **0.92±0.11** | **8.96±0.02** | **14.40±0.20** |
> |Do-PFN| 6.07±0.89 |  4.11±0.52 | 12.01±.03 | 20.91±0.14|
>
> **2) Mean ATE Relative Error ± Standard Error (↓ better)**
>
> | Method | IHDP | ACIC 2016 | Lalonde CPS | Lalonde PSID |
> |--|--|--|----|----|
> | **CausalPFN** | 0.20±0.04| 0.05±0.01 | **0.13±0.01** | **0.22±0.02** |
> |Do-PFN|0.57±0.10| 0.67±0.04| 0.87±0.01|0.92±0.01|
>
>
> As it is clear from both tables, CausalPFN achieves superior performance in all the benchmarks. We hypothesize that this is due to the non-identifiability of the prior data that they use, which we will elaborate in the next part.
>
>
> > Why do you only gather identifiable DAGs? Would you not want the distribution over all equivalent DAGs no matter if they are identifiable or not? This would probably give you less clean estimates but would that not better? I.e. right now the argument seems that you only care about a DAG if it is identifiable while in the real world that might not be the case.
>
> We have the following reasons to only include identifiable data-generating processes (DGP) in our prior generation:
>
> 1. Non-identifiability of the DGP means that for the _same observational distribution_, we can get an _unbounded set_ of causal effects. Therefore, if we allow for non-identifiable settings, in the worst case, we can have DGPs that all agree on the observational data (the only available information during inference) while having a range of $(-\infty, \infty)$ for the causal effects, which lacks any useful insights. Specifically, this worst-case scenario is more likely when generating large-scale prior data. This is perhaps one of the reasons we see worse performance for Do-PFN compared to our CausalPFN, as demonstrated in the previous part, since Do-PFN uses non-identifiable prior data, potentially having unobserved confounding.
>
>
> 2. More concretely, we have provided an updated version of Proposition 1 here, that states that **identifiability is necessary for convergence** of the posterior-predictive distribution of causal effects (the quantity that CausalPFN aims to learn) in the limit:
>
>
> **Proposition 1** (Informal). Under mild regularity assumptions, for almost all $\psi^\star \sim \pi$ and any set of i.i.d. samples
> $\mathcal{D}\_\text{obs}\sim\mathrm{P}^{\psi^\star}\_\text{obs}$, we have that as $|\mathcal{D}\_\text{obs}|\to\infty$,
>
> $$\begin{align}
>   \mathbb{E}^{\pi^{\mu\_t}} \bigl[\mu \mid \textbf{X},\mathcal{D}\_\text{obs}\\bigr]
>   \overset{a.s.}{\longrightarrow}
>   \mu_t(\textbf{X}; \psi^\star), \quad \forall t \in \mathcal{T},
>   \quad \textbf{X}\sim\mathrm{P}^{\psi^\star},
> \end{align}$$
>
> **if and only if** the support of the prior $\pi$ only contains $\psi$ with **identifiable** CEPOs almost everywhere.
>
> What this theory directly implies is that the identifiability (or in our case ignorability) requirement of the DGPs is not only sufficient for consistent estimation of the causal effects, but it is also a _necessity_. We have a formal proof for this proposition, which we will include in the final version of the paper. We would also be happy to provide the proof during the discussion period if requested.
>
> > Why do you only provide inference code and not training code?
>
> We will release the training code, along with the synthetic prior data used for the training, with the published version of the paper.

---

> > ### Comment · Reviewer_8PoL · 2025-08-01
> >
> > Thanks for the clarifications for mine as well as for other reviews. I have read all reviews and all responses. I agree with the clear majority of the reviewers that this work should be accepted.

---

### Official Review · Reviewer_drth · 2025-07-01

**Clarity:** 3
**Significance:** 3
**Originality:** 3
**Rating:** 5
**Confidence:** 4

**Summary:**

This paper introduces Causal Prior-Fitted Networks, an extension of PFNs to the problem of causal effect estimation. The authors reframe the PFN framework for causal tasks and sketch a proof of consistency of the estimates. The paper makes use of an efficient data generation approach relying on real world datasets and adding appropriate variables for causal tasks. The experiments show the versatility of the framework and overall outstanding performance compared to relevant baselines.

**Questions:**

- Does the PFN also capture data-dependent uncertainty? Ie. do you observe the model becoming more confident with more data in the observational dataset?
- What are the limits in terms of number of covariates? Data samples? ..?
- How did you ensure that your training datasets did not contain data that you tested on?
- Could this be extended into handling time series tasks?


- L58: “Fitted on the ignorability assumption” - How do you handle real-world situations where the data might not come from a DGP that fulfils this criterion?
- L158 “ATEs… averaging across units in D_obs” - How would this work?
- L 179: typo -> illustrated
- L192 / data generation: Is positivity really guaranteed? You could still generate datasets where on treatment is never sampled. Is this checked? What about edge cases where the random column chosen for Y0/1 is the copy of a column in X?
- How would this work for continuous treatments? I guess the method itself should work and it’s “only” a data generation issue?
- L207 How is the histogram handled for prediction? What are the bin locations? Is the data normalised to ensure equivalent bins across datasets?
- L227ff: Could you define the metrics properly? Why are relative errors directly averageable and PEHEs not? Why not use relative errors for CATE too?
- L243: You could still get a CATE on RCTs depending on treatment assignments? I guess here you’re talking about ITEs almost? Should be specific here.
- Eq 10: What’s \lambda (not \lambda(q))?
- Table2: How are the scores normalised?
- Table 3: Why does fine-tuning decrease its performance?
- A comparison to other causal zero-shot methods like [1] would have been great.

[1] Zhang, Jiaqi, et al. "Towards causal foundation model: on duality between causal inference and attention." arXiv preprint arXiv:2310.00809 (2023).

**Ethical Concerns:**

["NO or VERY MINOR ethics concerns only"]

**Limitations:**

mostly. see questions

**Quality:**

4

**Strengths And Weaknesses:**

+ The paper is well structured and written, making it easy to follow.
+ The paper offers a convincing and supposedly easy to use method for real-world causal effect estimation that has been evaluated on a wide range of synthetic, semi-synthethic and real tasks.
+ The data generation procedure is simple yet effective ensuring that the datasets are not bound to certain functional relationships between covariates and outcomes.
+ The evaluation touches upon interesting properties such as the calibration of the model.

- Some areas of the paper are lacking in clarity. E.g. the metrics should be better defined.
- Comparisons to other amortised causal inference methods are lacking and would further support this work.

---

> ### Author Rebuttal · Authors · 2025-07-31
>
> We thank the reviewer for their thoughtful review, insightful questions, and their positive assessment of the paper, as well as their view that CausalPFN is both convincing and easy to use.
>
> ### More Confident Model With More Data
>
> CausalPFN becomes more confident as it observes more data. As an example, we use the synthetic polynomial datasets described in L1262–L1275 of the appendix and provide the length of the 95% credible interval of CATE estimates from CausalPFN, along with the confidence interval derived from a DR-Learner:
>
> **Table: The Effect of Sample Size on the Length of the 95% Credible Interval**
>
> | Method\Sample Size | 10 | 20 | 50 | 100 | 200 | 500 | 1000 | 5000 | 10000 |
> |---|---|-|-|---|---|---|--|--|--|
> |**CausalPFN**|12.67±0.46|6.40±0.33|4.74±0.11|3.98±0.08|2.98±0.08|2.01±0.04|1.67±0.03|0.98±0.04|0.97±0.03|
> |DR-Learner|5.25±0.68|2.49±0.15|1.20±0.03|0.83±0.01|0.73±0.02|0.69±0.01|0.69±0.01|0.58±0.01|0.52±0.01|
>
> Although more conservative than the doubly-robust learner, the length of CausalPFN’s credible interval goes down as we observe more samples. Moreover, to ensure well-calibration, we also calculate the integrated coverage error (ICE):
>
> **CausalPFN’s ICE for Different Sample Sizes**
>
> |10  | 20| 50  | 100  | 200 | 500 | 1000 | 5000 | 10000 |
> |---|---|---|---|--|--|---|---|---|
> |0.13±0.01|0.01±0.01|0.01±0.01|0.04±0.00|0.02±0.01|0.05±0.00|0.03±0.00|-0.01±0.01|-0.04±0.01|
>
> ### The Limit of Covariate and Sample Size
>
> During the causal phase of the training, we consider sample sizes up to 2048 and covariates up to 99. However, during inference, CausalPFN can take $50{,}000$ (Table 2) or more (Table 5).
>
> To assess the effect of sample/covariate size on CausalPFN’s performance, we provide additional experiments. Similar to above, we focus on the synthetic polynomial datasets. We fix the test size at 100 samples in all experiments, and for each setting of sample size and covariate size, we report the average±standard error over 50 tables generated from the polynomial prior with a fixed random seed.
>
> **1) Effect of Sample Size**: Here, we consider the same data-generating process while increasing the number of samples. We fix the number of covariates to 10. The table below shows the PEHE values for CausalPFN’s CATE compared to other baselines. It is clear that CausalPFN has a better ability to utilize large sample sizes, as the PEHE drops faster.
>
> | Method\Sample Size | 10  | 20| 50 | 100 | 200 | 500 | 1000 | 5000  | 10000 |
> |--|--|---|--|---|--|---|---|----|--|
> | **CausalPFN**  | 1.34±0.02| 1.27±0.02| 1.10±0.02| 0.89±0.02| 0.74±0.03| 0.46±0.01| 0.40±0.01| 0.29±0.01| 0.31±0.01|
> | DA Learner| 1.33±0.02| 1.30±0.02| 1.16±0.01| 1.00±0.01| 0.91±0.03| 0.85±0.01| 0.85±0.02| 0.84±0.02| 0.82±0.02|
> | S Learner | 1.44±0.01| 1.40±0.02| 1.35±0.02| 1.21±0.02| 1.18±0.05| 1.07±0.03| 1.09±0.03| 1.00±0.04| 1.03±0.04|
> | T Learner | 1.33±0.02| 1.30±0.02| 1.15±0.01| 0.97±0.01| 0.88±0.02| 0.81±0.01| 0.83±0.01| 0.81±0.02| 0.81±0.02|
> | X Learner | 1.35±0.02| 1.32±0.02| 1.20±0.02| 1.04±0.01| 0.94±0.03| 0.87±0.01| 0.88±0.02| 0.87±0.02| 0.84±0.02|
>
> **2) Effect of Covariate Size**: Here, we fix the number of samples to 1000 and increase the number of covariates. Similar to above, the table below compares the performance of CausalPFN to other methods in terms of PEHE. Although CausalPFN consistently maintains a performance gap, the gap becomes smaller as the number of covariates increases. This is likely due to the limited number of covariates during training (up to 100) and can potentially be addressed by increasing the number of covariates during training.
>
> | Method\\# of Covariates | 1 | 5 | 10 | 20 | 50 | 100 | 200 | 500  | 1000 |
> |-|--|--|--|--|--|--|--|--|--|
> | **CausalPFN** | 0.08±0.00| 0.17±0.01| 0.40±0.01| 0.67±0.01| 0.87±0.02| 1.01±0.02| 1.17±0.02| 1.28±0.02| 1.32±0.02|
> | DA Learner| 0.21±0.01| 0.59±0.01| 0.85±0.01| 1.04±0.01| 1.14±0.01| 1.22±0.01| 1.25±0.01| 1.30±0.02| 1.32±0.02|
> | S Learner | 0.54±0.04| 0.83±0.02| 1.09±0.02| 1.17±0.02| 1.21±0.02| 1.23±0.01| 1.26±0.02| 1.30±0.02| 1.33±0.02|
> | T Learner| 0.27±0.01| 0.60±0.01| 0.83±0.01| 0.98±0.01| 1.09±0.01| 1.18±0.01| 1.23±0.02| 1.28±0.02| 1.32±0.02|
> | X Learner | 0.29±0.01| 0.64±0.01| 0.88±0.01| 1.06±0.01| 1.15±0.01| 1.23±0.01| 1.26±0.02| 1.30±0.02| 1.33±0.02|
>
> ### Ensuring Training Datasets Did Not Contain Test Data
>
> 1. Most of the pre-training data used in our model are purely synthetic, following a similar approach to TabPFN [42]. We will publish the training code, along with the pre-training data, for the final version.
>
> 2. For the real-world part of the pre-training data, we only use real-world covariates and generate the treatments through synthetic functions and add synthetic noise to the CEPO columns to generate potential outcomes.  We included the list of real-world datasets (OpenML datasets) in footnote 2 on page 28 of the appendix, which shows that none of the benchmarks are part of those datasets.
>
> ### Extension to Time Series Tasks
>
> Potentially, there have already been a few attempts to use in-context learners on time-series data [B, C].
>
> [B] Hoo, Shi Bin, et al. "From Tables to Time: How TabPFN-v2 Outperforms Specialized Time Series Forecasting Models." arXiv:2501.02945
>
> [C] Taga, Ege Onur, et al. "TimePFN: Effective multivariate time series forecasting with synthetic data." AAAI 2025.
>
> ### L58: The Ignorability Assumption
>
> CausalPFN is only applicable for the ignorability assumptions where identifiability holds. In most cases with no ignorability (i.e., with unobserved confounding), the causal effects can be generally unbounded. This is why many standard causal effect estimators are based on this assumption, and rely on a domain expert to assess whether ignorability holds.
>
> ### L158: ATE Calculation by Averaging Across Units in $\mathcal{D}\_\text{obs}$
>
> We run the model to estimate CATE for each observed unit and calculate the average of all to estimate the ATE.
>
> ### L192: Is Positivity Guaranteed?
>
> Positivity is guaranteed in the population as the treatment logits go through a Sigmoid function to generate propensities $P(T=1|X) \in (0, 1)$. Still, there is a small chance that the data will only contain one treatment due to finite samples. We explicitly discard those edge cases during training.
>
> ### Continuous Treatments
>
> In principle, the CEPO-PPDs are still well-defined and the causal data-prior loss should still yield valid CEPO-PPDs. The challenge here, as correctly mentioned by the reviewer, is to implement a diverse and smooth data generation pipeline.
>
>
> ### L207: Histogram Loss
> As provided in Section B.2 L1175-L1179 of the appendix, the data is normalized first, then 1024 equidistant bins are allocated from -10 to +10; every point outside of this [-10, +10] is treated as an outlier and clamped.
>
> Once the prediction is done in [-10, +10], we use the stored scale and shift parameter using for normalization to undo it and bring it back to the original scale.
>
> ### Metric Definitions
>
> For CATE evaluation, for a test dataset of size $N$ and ground-truth CATEs $\tau$ alongside their estimations $\hat{\tau}$, the evaluation metric, PEHE, is defined as:
>
> $$\text{PEHE}(\hat{\tau}) = \sqrt{\frac{1}{N} \sum_{n=1}^{N} (\hat{\tau}(\textbf{x}^{(n)}) - {\tau}(\textbf{x}^{(n)}))^2}$$
>
> For ATE evaluation, given a ground-truth ATE $\lambda$ and its estimation $\hat{\lambda}$, the relative error is defined as: $\frac{|\lambda - \hat{\lambda}|}{|\lambda|}$.
>
> We will include these equations in the revised version.
>
> > Why are relative errors directly averageable and PEHEs not?
>
> PEHE depends on the scale of CATE values. Hence, averaging across different datasets would be biased towards datasets with larger output values, while the scale of ATE is normalized in calculating the relative error. For more consistency, we will update Table 1 with average rank instead of average relative errors for ATE estimation, similar to CATE.
>
> > Why not use relative errors for CATE?
>
> PEHE is a standard metric used for CATE estimation in the causal inference literature. Moreover, the CATE error is defined as an average metric over $N$ samples, while the ATE error is defined only for one value per dataset.
>
> ### L243: ITE vs CATE
>
> We consider CATE, but the conditioning variables consist of all the observed covariates. In that sense, it is close to ITEs (though we still estimate an expected value). We will clarify this in the paper.
>
> ### Eq 10: What is $ \lambda$?
>
> $\lambda$ coincides with ATE in equation (2) (same notation). We will recall its definition for more clarity.
>
> ### Table2: Score Normalization
>
> We normalize each column by dividing all values by the maximum score in that column. This normalization enables us to compute an average performance metric across datasets
>
> ### Table 3: Fine-tuning vs. Performance
>
> In Table 3, the first row corresponds to CausalPFN, while the second and third rows report results for a different underlying architecture (TabPFNv2): one without fine-tuning (row 3) and one with fine-tuning on our causal prior (row 2). In all cases, fine-tuning improves performance when comparing the second and third rows.
>
> ### Comparison to "Towards causal foundation model”
>
> We have already briefly discussed this method (see [110]). Unfortunately, we could not find any implementation for this method. Moreover, as mentioned in Section 5.3 in [110], the zero-shot version of their method, CInA (ZS), is only applicable for benchmarks that naturally come with multiple datasets and there is no clear instruction on how to use the zero-shot version for standard casual effect estimation tasks akin to the ones we have used.

---

> > ### Comment · Area_Chair_egC9 · 2025-08-06
> >
> > Dear drth,
> >
> > Thank you again for your review. As we approach the final decision, please take a moment to engage with the authors’ rebuttal—either by noting any remaining concerns or by explicitly acknowledging it if your overall assessment is unchanged.
> > Appreciate your time and input.
> >
> > Best,
> > AC

---

### Official Review · Reviewer_MX9h · 2025-07-03

**Clarity:** 3
**Significance:** 3
**Originality:** 3
**Rating:** 5
**Confidence:** 3

**Summary:**

The paper introduces a new causal effect estimation method. The Authors leverage prior-fitted networks to amortize the estimation effort and increase both accuracy and efficiency. The method is theoretically justified to work under a strong ignorability assumption. Empirical results across multiple benchmarks demonstrate that the proposed approach performs competitively compared to a diverse set of baselines.

**Questions:**

1. How large are the datasets used in terms of sample size and number of covariates?  How many covariates are considered both during training and evaluation? How does the method’s performance in terms of accuracy scale with the number of covariates compared to other methods?
2. Could you provide a version of Figure 1 that isolates inference time only, excluding training and hyperparameter search for the baseline methods? This would help in fairly assessing the computational efficiency of CausalPFN

**Ethical Concerns:**

["NO or VERY MINOR ethics concerns only"]

**Final Justification:**

The rebuttal assured me of the paper's quality. The additional experiments have resolved my concerns. I decided to keep my score.

**Limitations:**

yes

**Quality:**

3

**Strengths And Weaknesses:**

**Strengths**:
1. The presented approach is novel, interesting, and well motivated by recent advances in generative modelling and the challenges in causal inference.
2. The paper is well written, and the core ideas are presented clearly and convincingly.
3. The experimental evaluation is extensive and sound.

**Weaknesses**:
1. Details regarding method scalability are not discussed. (see question 1)

---

> ### Author Rebuttal · Authors · 2025-07-31
>
> We thank the reviewer for their effort and considering our paper as novel, well-motivated, clearly explained, and our experiments as extensive. In what follows, we have provided detailed responses to each individual comment:
>
> > Details regarding method scalability are not discussed. How large are the datasets used in terms of sample size and number of covariates? How many covariates are considered both during training and evaluation?
>
> In the appendix (footnote 2 on page 28), we provided an anonymized link to the list of all the real-world datasets (OpenML tables), which includes details such as sample sizes and the number of features. For the causal phase of the training (synthetic datasets), the following table summarizes the characteristics of the tables used, which we will include in our revision.
>
> **Table: Description of Training Data**
> | Description    | Value  |
> |----|----|
> | Number of Datasets   |  $\approx 11{,}140{,}000$   |
> | Range of Sample Sizes in the Context (Training)   | $(\frac13 \times 2048, \frac23 \times 2048)$  |
> | Range of Sample Sizes in the Query (Test)   | $(\frac13 \times 2048, \frac23 \times 2048)$  |
> | Range of the Number of Covariates   | $[1,99]$   |
>
> For evaluation, we used standard benchmarks, and we detail the number of samples and covariates below:
>
> **Table: Description of Benchmarks**
> | Dataset   | Training Sample Size| Test Sample Size |Number of Covariates  |
> |---|----|----|--|
> | IHDP  |  $672$  | $75$ | $25$ |
> | ACIC 2016  | $4{,}321$  | $481$ | $58$ |
> | Realcause Lalonde CPS  | $14{,}559$ | $1{,}618$ | $8$ |
> | Realcause Lalonde PSID  | $2{,}407$  | $268$ | $8$ |
>
> For the marketing datasets, as stated in the paper, we perform 5-fold honest splitting. We evaluate both the original sizes (Table 5) and a stratified 50k sub-sampling (Table 2).
>
> **Table: Description of Benchmarks**
> | Dataset | Sample Size| Number of Covariates   |
> |----|----|---|
> |Hill (1)  | $42{,}963$ | $10$ |
> |Hill (2) | $42{,}613$ | $10$ |
> |Criteo | $1{,}397{,}960$ | $12$ |
> |Lenta | $687{,}029$ | $193$ |
> |X5 |$200{,}039$|$6$|
> |Megafon|$600{,}000$|$50$|
>
> To handle more than 99 covariates, we apply PCA to the covariates and retain only the first 99 principal components to obtain a condensed representation of the covariates that can fit into the input of the model.
>
> > How does the method’s performance in terms of accuracy scale with the number of covariates compared to other methods?
>
> Based on your suggestion, we conducted an additional ablation experiment to assess the effect of the number of covariates on the performance of CausalPFN compared to other methods. For this ablation experiment, we focus on the synthetic polynomial datasets described in lines (1262–1275) of the appendix. We fix the test size at 100 samples and number of training samples as 1000, and increase the number of covariates. For each covariate size, we report the average ± standard error of PEHE of CATE estimates over 50 tables generated from the polynomial prior with a fixed random seed to control for randomness:
>
> **Table: Effect of Covariate Size**
> | Method\\# of Covariates| $1$  | $5$ | $10$ | $20$ | $50$  | $100$ | $200$ | $500$  | $1000$  |
> |----|--|---|----|---|----|-----|---|---|--|
> | **CausalPFN**  | $0.08±0.00$| $0.17±0.01$| $0.40±0.01$| $0.67±0.01$| $0.87±0.02$| $1.01±0.02$| $1.17±0.02$| $1.28±0.02$| $1.32±0.02$|
> | DA Learner| $0.21±0.01$| $0.59±0.01$| $0.85±0.01$| $1.04±0.01$| $1.14±0.01$| $1.22±0.01$| $1.25±0.01$| $1.30±0.02$| $1.32±0.02$|
> | S Learner| $0.54±0.04$| $0.83±0.02$| $1.09±0.02$| $1.17±0.02$| $1.21±0.02$| $1.23±0.01$| $1.26±0.02$| $1.30±0.02$| $1.33±0.02$|
> | T Learner| $0.27±0.01$| $0.60±0.01$| $0.83±0.01$| $0.98±0.01$| $1.09±0.01$| $1.18±0.01$| $1.23±0.02$| $1.28±0.02$| $1.32±0.02$|
> | X Learner| $0.29±0.01$| $0.64±0.01$| $0.88±0.01$| $1.06±0.01$| $1.15±0.01$| $1.23±0.01$| $1.26±0.02$| $1.30±0.02$| $1.33±0.02$|
>
> The table suggests that CausalPFN consistently maintains a performance gap over different numbers of covariates. However, the gap becomes smaller as the number of covariates increases. This is likely due to the limited number of covariates during training (up to 100). We believe that increasing the number of covariates during training can address this issue. Still, we observe that CausalPFN can generalize well to a larger number of covariates.
>
>
> > Could you provide a version of Figure 1 that isolates inference time only, excluding training and hyperparameter search for the baseline methods? This would help in fairly assessing the computational efficiency of CausalPFN.
>
>
> We included the training and hyperparameter tuning time for other baselines in Figure 1 as this is what would normally happen in practice: when faced with a causal task, domain experts typically search over a set of possible design choices, including different hyperparameters. On the other hand, CausalPFN is only pre-trained once, and for each new dataset, it only requires running a few forward passes of the transformer model. We want to emphasize that Figure 1 is not intended to claim that the inference speed of CausalPFN is inherently faster than that of the baselines; rather, it illustrates what a practitioner would typically experience in terms of end-to-end runtime when choosing an approach.
>
> Having said that, we agree with the reviewer that providing inference-only time can provide more insights into the efficiency of CausalPFN. However, including the inference-only time requires re-training all the baselines, which can take more than a week of computation and is infeasible for the rebuttal period. Instead, we provide results that we already had available that contain the training + inference time for the baselines, excluding the hyperparameter optimization time.
>
> **Table: Median Time for CATE Estimation (Seconds per 1000 Samples)**
>
> | Method                  | ACIC 2016 | IHDP     | RealCause CPS | RealCause PSID | Overall   |
> |------------------------|-----------|----------|----------------|----------------|-----------|
> | IPW           | `0.059753` | `0.125096` | `0.040139`     | `0.081708`     | `0.078507` |
> | S Learner    | `0.075550` | `0.454180` | `0.021157`     | `0.115120`     | `0.118159` |
> | T Learner    | `0.116674` | `0.883767` | `0.043567`     | `0.245745`     | `0.247548` |
> | DA Learner   | `0.283248` | `2.076818` | `0.089609`     | `0.535804`     | `0.548800` |
> | **CausalPFN**             | `0.630637` | `0.673687` | `0.818941`     | `0.485863`     | `0.555087` |
> | GRF          | `1.051030` | `0.596345` | `0.708011`     | `0.510705`     | `0.579403` |
> | X Learner    | `0.328857` | `2.585276` | `0.105298`     | `0.691134`     | `0.679522` |
> | TarNet       | `1.293894` | `5.866852` | `0.314323`     | `1.102849`     | `1.053019` |
> | DragonNet    | `1.506751` | `4.659946` | `1.168565`     | `2.607162`     | `1.916093` |
> | RA Net       | `4.501656` | `14.892828`| `1.129872`     | `4.681562`     | `3.883391` |
> | Forest DR Learner     | `424.962053`| `25.474570`| `2.920928`    | `3.989793`     | `3.892557` |
> | Forest DML   | `554.478033`| `32.682874`| `3.836490`    | `5.007255`     | `5.222570` |
> | BART                  | `4.509301` | `5.668311` | `9.506232`     | `9.264590`     | `8.637354` |

---

> > ### Comment · Reviewer_MX9h · 2025-08-04
> >
> > I want to thank the Authors for the detailed response and additional experimental results. The provided experiments and explanations resolve my concerns.

---

### Note · Authors · 2025-08-12

We appreciate the reviewers' thorough evaluation and constructive engagement - we’re grateful for all the feedback received and have updated the manuscript with the same.
All reviewers found our work **novel**, **well-motivated**, and **technically sound with extensive experiments**. The initial scores were already positive, and after our rebuttal, reviewers MX9h’s and 8PoL’s concerns were fully resolved and Reviewer URzh raised their score after we addressed the baseline comparisons.

**(Key highlights about our paper)**

1. **Pushing the boundary in amortized causal inference:** CausalPFN is the first successful use of Prior-Data Fitted Networks to causal effect estimation, achieving **zero-shot superior performance** across all major benchmarks (including real-world marketing data) while being orders of magnitude faster than traditional methods.

2. **Immediate practical impact:** Unlike existing methods requiring extensive hyperparameter tuning, CausalPFN works out-of-the-box, democratizing access to high-quality causal inference for practitioners. The reviewers recognized this as "a tool which hopefully can make it into the toolbox of practitioners" (8PoL).

3. **Theoretical foundation:** We provided rigorous theoretical justification (Proposition 1) showing identifiability is both necessary and sufficient for convergence.

4. **Comprehensive evaluation:** Our experiments span 9 benchmarks and compare against 15 baselines (including newly added methods like FlexTENet/SNet per URzh’s request), consistently demonstrating superior performance.

The reviewers' consensus that this work represents a significant advance in causal inference suggests strong potential for broader community interest. As 8PoL notes, this opens up a new paradigm that others can "build on top e.g. including interventions, other treatment forms etc."

Finally, we've committed to releasing all training code and data with the camera-ready version, ensuring full reproducibility.

Thank you for your consideration of this work.

Best,
Authors

---

### Decision · Program_Chairs · 2025-09-17

**Decision:**

Accept (spotlight)

**Comment:**

This paper introduces CausalPFN, a novel framework that amortizes causal effect estimation via prior-fitted networks and in-context learning. The method achieves strong zero-shot performance across standard benchmarks, is theoretically well grounded, and demonstrates practical utility by offering an out-of-the-box estimator without further tuning. Reviewers consistently found the paper novel, technically solid, and impactful, and all recommended acceptance.

The main points for improvement are relatively minor and can be addressed in the camera-ready version:
* Expand the discussion of very recent related works (e.g., concurrent PFN-based causal inference approaches) to better situate the contribution.
* Incorporate additional clarifications on scalability that were provided during rebuttal.
* Emphasize practical scenarios where computational speed is particularly beneficial.
* Ensure full release of both inference and training code, along with synthetic prior data, to guarantee reproducibility.